# Sampling from Gaussian Process Posteriors using Stochastic Gradient Descent

**Jihao Andreas Lin**[*1,2]    **Javier Antorán**[*1]    **Shreyas Padhy**[*1]
**David Janz**[3]    **José Miguel Hernández-Lobato**[1]    **Alexander Terenin**[1,4]
[1]University of Cambridge    [2]Max Planck Institute for Intelligent Systems
[3]University of Alberta    [4]Cornell University

## Abstract

Gaussian processes are a powerful framework for quantifying uncertainty and for sequential decision-making but are limited by the requirement of solving linear systems. In general, this has a cubic cost in dataset size and is sensitive to conditioning. We explore stochastic gradient algorithms as a computationally efficient method of approximately solving these linear systems: we develop low-variance optimization objectives for sampling from the posterior and extend these to inducing points. Counterintuitively, stochastic gradient descent often produces accurate predictions, even in cases where it does not converge quickly to the optimum. We explain this through a spectral characterization of the implicit bias from non-convergence. We show that stochastic gradient descent produces predictive distributions close to the true posterior both in regions with sufficient data coverage, and in regions sufficiently far away from the data. Experimentally, stochastic gradient descent achieves state-of-the-art performance on sufficiently large-scale or ill-conditioned regression tasks. Its uncertainty estimates match the performance of significantly more expensive baselines on a large-scale Bayesian optimization task.

## 1 Introduction

Gaussian processes (GPs) provide a comprehensive framework for learning unknown functions in an uncertainty-aware manner. This often makes Gaussian processes the model of choice for sequential decision-making, achieving state-of-the-art performance in tasks such as optimizing molecules in computational chemistry settings [21] and automated hyperparameter tuning [39, 24].

The main limitations of Gaussian processes is that their computational cost is cubic in the training dataset size. Significant research efforts have been directed at addressing this limitation resulting in two key classes of scalable inference methods: (i) *inducing point* methods [43, 23], which approximate the GP posterior, and (ii) *conjugate gradient* methods [20, 19, 4], which approximate the computation needed to obtain the GP posterior. Note that in structured settings, such as geospatial learning in low dimensions, specialized techniques are available [49, 48, 15]. Throughout this work, we focus on the generic setting, where scalability limitations are as of yet unresolved.

In recent years, stochastic gradient descent (SGD) has emerged as the leading technique for training machine learning models at scale [37, 42], in both deep learning and related settings like kernel methods [14] and Bayesian modeling [29]. While the principles behind the effectiveness of SGD are not yet fully understood, empirically, SGD often leads to good predictive performance—even when it does not fully converge. The latter is the default regime in deep learning, and has motivated researchers to study *implicit biases* and related properties of SGD [6, 52].

---

[*]Equal contribution, order chosen randomly.
Code available at: HTTPS://GITHUB.COM/CAMBRIDGE-MLG/SGD-GP.

37th Conference on Neural Information Processing Systems (NeurIPS 2023).

In the context of GPs, SGD is commonly used to learn kernel hyperparameters—by optimizing the marginal likelihood [18, 13, 19, 12, 11] or closely related variational objectives [43, 23]. In this work, we explore applying SGD to the complementary problem of approximating GP posterior samples given fixed kernel hyperparameters. In one of his seminal books on statistical learning theory, Vladimir Vapnik [45] famously said: *"When solving a given problem, try to avoid solving a more general problem as an intermediate step."* Motivated by this viewpoint, as well as the aforementioned property of good performance often not requiring full convergence when using SGD, we ask: *Do the linear systems arising in GP computations necessarily need to be solved to a small error tolerance? If not, can SGD help accelerate these computations?*

We answer the latter question affirmatively, with specific contributions as follows: (i) We develop a scheme for drawing GP posterior samples by applying SGD to a quadratic problem. In particular, we re-cast the pathwise conditioning technique of Wilson et al. [50, 51] as an optimization problem to which we apply the low-variance SGD sampling estimator of Antorán et al. [3]. We extend the proposed method to inducing point Gaussian processes. (ii) We characterize the implicit bias in SGD-approximated GP posteriors showing that despite optimization not fully converging, these match the true posterior in regions both near and far away from the data. (iii) Experimentally, we show that SGD produces strong results—compared to variational and conjugate gradient methods—on both large-scale and poorly-conditioned regression tasks, and on a parallel Thompson sampling task, where error bar calibration is paramount.

## 2   Optimization and Pathwise Conditioning in Gaussian Processes

A random function $f : X \to \mathbb{R}$ over some set $X$ is a *Gaussian process* [35] if, for every finite set of points $\boldsymbol{x} \in X^N$, $f(\boldsymbol{x})$ is multivariate Gaussian. A Gaussian process $f \sim \mathrm{GP}(\mu, k)$ is uniquely determined by a *mean function* $\mu(\cdot) = \mathbb{E}(f(\cdot))$ and a *covariance kernel* $k(\cdot, \cdot') = \mathrm{Cov}(f(\cdot), f(\cdot'))$. We denote by $\mathbf{K}_{\boldsymbol{xx}}$ the *kernel matrix* $[k(x_i, x_j)]_{i,j=1,\dots,N}$. We consider the Bayesian model $\boldsymbol{y} = f(\boldsymbol{x}) + \boldsymbol{\varepsilon}$, where $\boldsymbol{\varepsilon} \sim \mathrm{N}(\boldsymbol{0}, \boldsymbol{\Sigma})$ gives the likelihood, $f \sim \mathrm{GP}(0, k)$ is the prior, and $\boldsymbol{x}, \boldsymbol{y}$ are the training data. The posterior of this model is $f \mid \boldsymbol{y} \sim \mathrm{GP}(\mu_{f|\boldsymbol{y}}, k_{f|\boldsymbol{y}})$, with

$$\mu_{f|\boldsymbol{y}}(\cdot) = \mathbf{K}_{(\cdot)\boldsymbol{x}}(\mathbf{K}_{\boldsymbol{xx}} + \boldsymbol{\Sigma})^{-1}\boldsymbol{y} \qquad k_{f|\boldsymbol{y}}(\cdot, \cdot') = \mathbf{K}_{(\cdot, \cdot')} - \mathbf{K}_{(\cdot)\boldsymbol{x}}(\mathbf{K}_{\boldsymbol{xx}} + \boldsymbol{\Sigma})^{-1}\mathbf{K}_{\boldsymbol{x}(\cdot')}. \quad (1)$$

Using pathwise conditioning [50, 51] one can also write the posterior directly as the random function

$$(f \mid \boldsymbol{y})(\cdot) = f(\cdot) + \mathbf{K}_{(\cdot)\boldsymbol{x}}(\mathbf{K}_{\boldsymbol{xx}} + \boldsymbol{\Sigma})^{-1}(\boldsymbol{y} - f(\boldsymbol{x}) - \boldsymbol{\varepsilon}) \quad \boldsymbol{\varepsilon} \sim \mathrm{N}(\boldsymbol{0}, \boldsymbol{\Sigma}) \quad f \sim \mathrm{GP}(0, k). \quad (2)$$

We assume throughout that $\boldsymbol{\Sigma}$ is diagonal. Due to the matrix inverses present, the cost to directly compute each of the above expressions is $\mathcal{O}(N^3)$.

### 2.1   Random Fourier Features and Efficient Sampling

Our techniques will rely on *random Fourier features* [34, 40]. Let $X = \mathbb{R}^d$ and let $k$ be stationary—that is, $k(x, x') = k(x - x')$. Assume, in this section only and without loss of generality, that $k(x, x) = 1$. Random Fourier features are sets of $L$ random functions $\boldsymbol{\Phi} : X \to \mathbb{R}^L$ whose components, indexed by $\ell \leq L$, are $\phi_\ell(\cdot) = L^{-1/2}\cos(2\pi\langle\omega_\ell, \cdot\rangle)$ for $\ell$ odd, and $\phi_\ell(\cdot) = L^{-1/2}\sin(2\pi\langle\omega_\ell, \cdot\rangle)$ for $\ell$ even. By taking $\omega_\ell$ to be samples distributed according to $\rho$, the normalized spectral measure of the kernel, for any $x, x' \in X$ we have

$$k(x, x') = \mathbb{E}_{\omega_1, \dots, \omega_L \sim \rho}\langle\boldsymbol{\Phi}(x), \boldsymbol{\Phi}(x')\rangle, \quad (3)$$

which we will use to construct unbiased estimators of kernel matrices in the sequel. Fourier features can also be used to *efficiently sample* functions from the GP prior. Naively, evaluating the random function $f$ at $N_*$ points requires computing a matrix square root of $\mathbf{K}_{\boldsymbol{xx}}$ at $\mathcal{O}(N_*^3)$ computational cost. However, given a precomputed set of Fourier features $\boldsymbol{\Phi}$, the Monte Carlo estimator

$$f(\cdot) \approx \tilde{f}(\cdot) = \boldsymbol{\theta}^T \boldsymbol{\Phi}(\cdot) = \sum_{\ell=1}^{L} \theta_\ell \phi_\ell(\cdot) \qquad\qquad \boldsymbol{\theta} \sim \mathrm{N}(\boldsymbol{0}, \mathbf{I}) \quad (4)$$

approximately samples from $\mathrm{GP}(0, k)$ at $\mathcal{O}(N_*)$ cost. The approximation error in (4) decays as $L$ goes to infinity. Following Wilson et al. [50, 51], our proposed algorithms will approximately sample from the posterior by replacing $f$ with $\tilde{f}$ in the pathwise conditioning formula (2).

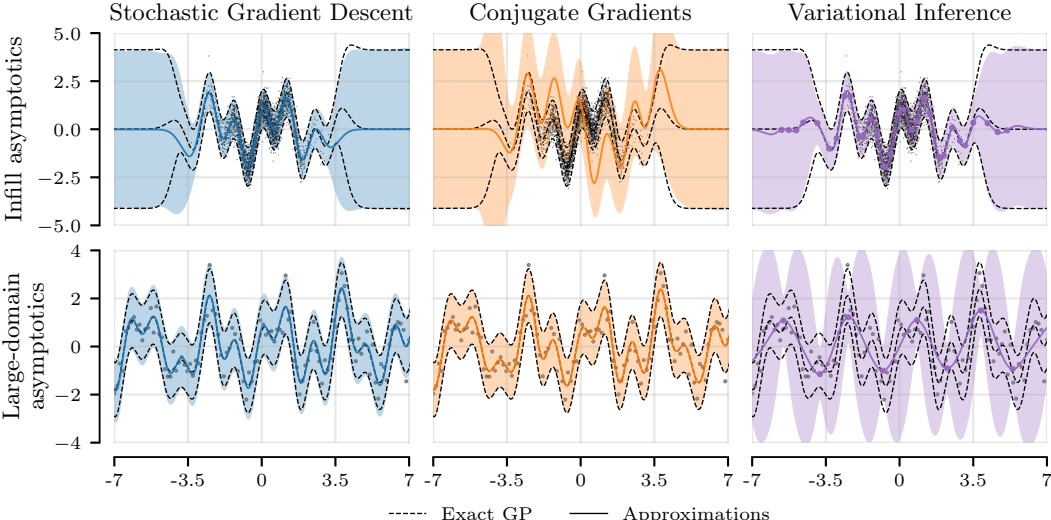

Figure 1: Comparison of SGD, CG [46] and SVGP [23] for GP inference with a squared exponential kernel on 10k datapoints from $\sin(2x) + \cos(5x)$ with observation noise $\mathrm{N}(0, 0.5)$. We draw 2000 function samples with all methods by running them for 10 minutes on an RTX 2070 GPU. *Infill asymptotics* considers $x_i \sim \mathrm{N}(0, 1)$. A large number of points near zero result in a very ill-conditioned kernel matrix, preventing CG from converging. SGD converges in all of input space except at the edges of the data. SVGP can summarise the data with only 20 inducing points. Note that CG converges to the exact solution if one uses more compute, but produces significant errors if stopped too early, as occurs under the given compute budget. *Large domain asymptotics* considers data on a regular grid with fixed spacing. This problem is better conditioned, allowing SGD and CG to recover the exact solution. However, 1024 inducing points are not enough for SVGP to summarize the data.

## 2.2 Optimization-based Learning in Gaussian Processes

Following Matthews et al. [31] and Antorán et al. [3], both a Gaussian process' posterior mean and posterior samples can be expressed as solutions to quadratic optimization problems. Letting $\boldsymbol{v}^* = (\mathbf{K}_{\boldsymbol{xx}} + \boldsymbol{\Sigma})^{-1}\boldsymbol{y}$ in (1), we can express the GP posterior mean in terms of the quadratic problem

$$\mu_{f|\boldsymbol{y}}(\cdot) = \mathbf{K}_{(\cdot)\boldsymbol{x}}\boldsymbol{v}^* = \sum_{i=1}^N v_i^* k(x_i, \cdot) \qquad \boldsymbol{v}^* = \arg\min_{\boldsymbol{v} \in \mathbb{R}^N} \sum_{i=1}^N \frac{(y_i - \mathbf{K}_{x_i\boldsymbol{x}}\boldsymbol{v})^2}{\Sigma_{ii}} + \|\boldsymbol{v}\|_{\mathbf{K}_{\boldsymbol{xx}}}^2. \quad (5)$$

We say that $k(x_i, \cdot)$ are the *canonical basis functions*, $v_i$ are the *representer weights* [38], and that $\|\boldsymbol{v}\|_{\mathbf{K}_{\boldsymbol{xx}}}^2 = \boldsymbol{v}^T\mathbf{K}_{\boldsymbol{xx}}\boldsymbol{v}$ is the *regularizer*. The respective optimization problem for obtaining posterior samples is similar, but involves a stochastic objective, which will be given and analyzed in Section 3.2.

*Conjugate gradients* (CG) are the most widely used algorithm to solve quadratic problems, both in the context of GPs [20, 19, 4] and more generally [33, 10]. CG reduces GP posterior inference to a series of matrix-vector products, each of $\mathcal{O}(N^2)$ cost. Given the system $\mathbf{A}^{-1}\boldsymbol{b}$, the number of matrix-vector products needed to guarantee convergence of CG to within a tolerance of $\varepsilon$ [41], is

$$\mathcal{O}\left(\sqrt{\mathrm{cond}(\mathbf{A})}\log\frac{\mathrm{cond}(\mathbf{A})\|\boldsymbol{b}\|}{\varepsilon}\right) \qquad\qquad \mathrm{cond}(\mathbf{A}) = \frac{\lambda_{\max}(\mathbf{A})}{\lambda_{\min}(\mathbf{A})}, \quad (6)$$

where $\lambda_{\max}(\mathbf{A})$ and $\lambda_{\min}(\mathbf{A})$ are the maximum and minimum eigenvalues of $\mathbf{A}$. CG performs well in many GP use cases, for instance Gardner et al. [19] and Wang et al. [46]. In general, however, the condition number $\mathrm{cond}(\mathbf{K}_{\boldsymbol{xx}} + \boldsymbol{\Sigma})$ need not be bounded, and conjugate gradients may fail to converge quickly [41]. Nonetheless, by exploiting the quadratic structure of the objective, CG obtains substantially better convergence rates than gradient descent [8, 52]. This presents a pessimistic outlook for gradient descent and related methods such as SGD which do not take explicit advantage of the quadratic nature of the objective. Nonetheless, the success of SGD in large-scale machine learning [9] motivates us to try it anyway.

# 3 Gaussian Process Predictions via Stochastic Gradient Descent

We now develop and analyze techniques for drawing samples from GP posteriors using SGD. This is done by rewriting the pathwise conditioning formula (2) in terms of two stochastic optimization problems. As a preview of what this will produce, we showcase SGD's performance on a pair of toy problems, designed to capture complementary computational difficulties, in Figure 1.

## 3.1 A Stochastic Objective for Computing the Posterior Mean

We begin by deriving a stochastic objective for the posterior mean. The optimization problem (5), with optimal representer weight solution $v^*$, requires $\mathcal{O}(N^2)$ operations to compute both its square error and regularizer terms exactly. The square error loss term is amenable to minibatching, which gives an unbiased estimate in $\mathcal{O}(N)$ operations. Assuming that $k$ is stationary, following Section 2.1, we can stochastically estimate the regularizer with Fourier features using the identity $\|v\|^2_{\mathbf{K}_{xx}} = \mathbb{E}_{\omega_1,\dots,\omega_L \sim \rho}\, v^T \mathbf{\Phi}^T(x)\mathbf{\Phi}(x)v$. Combining both estimators gives our SGD objective

$$\frac{N}{D}\sum_{i=1}^{D}\frac{(y_i - \mathbf{K}_{x_i x}v)^2}{\Sigma_{ii}} + \sum_{\ell=1}^{L}\left(v^T\phi_\ell(x)\right)^2 \tag{7}$$

where $D$ is the minibatch size and $L$ the number of Fourier features. This regularizer estimate is unbiased even when drawing a single Fourier feature per step: the number of features controls the variance. Equation (7) presents $\mathcal{O}(N)$ complexity, in contrast with the $\mathcal{O}(N^2)$ complexity of one CG step. We discuss sublinear inducing point techniques further on, but first turn to sampling.

## 3.2 Computing Posterior Samples

We now frame GP posterior samples in a manner amenable to SGD computation similarly to (7). First, we re-write the pathwise conditioning expression given in (2) as

$$(f \mid y)(\cdot) = \underbrace{f(\cdot)}_{\text{prior}} + \underbrace{\mu_{f\mid y}(\cdot)}_{\text{posterior mean}} - \underbrace{\mathbf{K}_{(\cdot)x}(\mathbf{K}_{xx} + \mathbf{\Sigma})^{-1}(f(x) + \varepsilon)}_{\text{uncertainty reduction term}} \quad \varepsilon \sim \mathrm{N}(\mathbf{0}, \mathbf{\Sigma}) \quad f \sim \mathrm{GP}(0, k). \tag{8}$$

We approximate the prior function sample using the Fourier feature approach of (4) and the posterior mean, defined in Section 2.2, with the minimizer $v^*$ of (7) obtained by SGD. Each posterior sample's uncertainty reduction term is parametrized by a set of representer weights given by a linear solve against a noisy prior sample evaluated at the observed inputs, namely $\alpha^* = (\mathbf{K}_{xx} + \mathbf{\Sigma})^{-1}(f(x) + \varepsilon)$. We construct an optimization objective targeting a sample's optimal representer weights as

$$\alpha^* = \underset{\alpha \in \mathbb{R}^N}{\arg\min}\sum_{i=1}^{N}\frac{(f(x_i) + \varepsilon_i - \mathbf{K}_{x_i x}\alpha)^2}{\Sigma_{ii}} + \|\alpha\|^2_{\mathbf{K}_{xx}} \qquad \begin{aligned} f(x) &\sim \mathrm{N}(\mathbf{0}, \mathbf{K}_{xx}) \\ \varepsilon &\sim \mathrm{N}(\mathbf{0}, \mathbf{\Sigma}). \end{aligned} \tag{9}$$

Applying minibatch estimation to this objective results in high gradient variance, since the presence of $\varepsilon_i$ makes the targets noisy. To avoid this while targeting the same objective, we modify (9) as

$$\alpha^* = \underset{\alpha \in \mathbb{R}^N}{\arg\min}\sum_{i=1}^{N}\frac{(f(x_i) - \mathbf{K}_{x_i x}\alpha)^2}{\Sigma_{ii}} + \|\alpha - \delta\|^2_{\mathbf{K}_{xx}} \qquad \begin{aligned} f(x) &\sim \mathrm{N}(\mathbf{0}, \mathbf{K}_{xx}) \\ \delta &\sim \mathrm{N}(\mathbf{0}, \mathbf{\Sigma}^{-1}), \end{aligned} \tag{10}$$

moving the noise term into the regularizer. This modification *preserves the optimal representer weights* since objective (10) equals (9) up to a constant: a proof is given in Appendix D. This generalizes the variance reduction technique of Antorán et al. [3] to the GP setting. Figure 2 illustrates minibatch gradient variance for these objectives. Applying the minibatch and random feature estimators of (7), we obtain a per-step cost of $\mathcal{O}(NS)$, for $S$ the number of posterior samples.

## 3.3 Inducing Points

So far, our sampling objectives have presented linear cost in dataset size. In the large-scale setting, algorithms with costs independent of the dataset size are often preferable. For GPs, this can be achieved through *inducing point posteriors* [43, 23], to which we now extend SGD sampling.

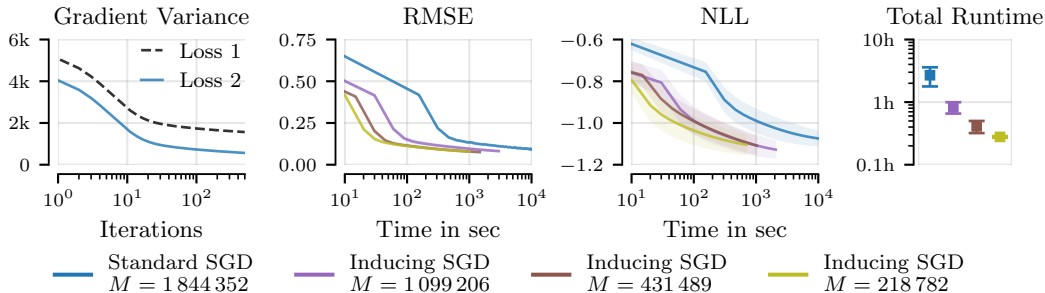

Figure 2: Left: gradient variance throughout optimization for a single-sample minibatch estimator ($D = 1$) of (9), labeled *Loss 1*, and the proposed sampling objective (10), labeled *Loss 2*, on the ELEVATORS dataset ($N \approx 16$k). Middle plots: test RMSE and negative log-likelihood (NLL) obtained by SGD and its inducing point variants, for decreasing numbers of inducing points, given in the rightmost plot, as a function of time on an A100 GPU, on the HOUSEELECTRIC dataset ($N \approx 2$M).

Let $\boldsymbol{z} \in X^M$ be a set of $M \in \mathbb{N}$ inducing points. Applying pathwise conditioning to the Kullback–Leibler-optimal inducing point approximation of Titsias [43] gives the expression

$$(f^{(\boldsymbol{z})} \mid \boldsymbol{y})(\cdot) = f(\cdot) + \mu_{f|\boldsymbol{y}}^{(\boldsymbol{z})}(\cdot) - \mathbf{K}_{(\cdot)\boldsymbol{z}}\mathbf{K}_{\boldsymbol{zz}}^{-1}\mathbf{K}_{\boldsymbol{zx}}(\mathbf{K}_{\boldsymbol{xz}}\mathbf{K}_{\boldsymbol{zz}}^{-1}\mathbf{K}_{\boldsymbol{zx}} + \boldsymbol{\Sigma})^{-1}(f^{(\boldsymbol{z})}(\boldsymbol{x}) + \boldsymbol{\varepsilon})$$
$$\boldsymbol{\varepsilon} \sim \mathrm{N}(\boldsymbol{0}, \boldsymbol{\Sigma}) \qquad f \sim \mathrm{GP}(0, k) \qquad f^{(\boldsymbol{z})}(\cdot) = \mathbf{K}_{(\cdot)\boldsymbol{z}}\mathbf{K}_{\boldsymbol{zz}}^{-1}f(\boldsymbol{z}). \tag{11}$$

Following Wild et al. [47], Theorem 5, the optimal inducing point mean $\mu_{f|\boldsymbol{y}}^{(\boldsymbol{z})}$ can therefore be written

$$\mu_{f|\boldsymbol{y}}^{(\boldsymbol{z})}(\cdot) = \mathbf{K}_{(\cdot)\boldsymbol{z}}\boldsymbol{v}^* = \sum_{j=1}^{M} v_i^* k(z_j, \cdot) \qquad \boldsymbol{v}^* = \underset{\boldsymbol{v} \in \mathbb{R}^M}{\arg\min} \sum_{i=1}^{N} \frac{(y_i - \mathbf{K}_{x_i\boldsymbol{z}}\boldsymbol{v})^2}{\Sigma_{ii}} + \|\boldsymbol{v}\|_{\mathbf{K}_{\boldsymbol{zz}}}^2 \tag{12}$$

and we can again parameterize the uncertainty reduction term (8) as $\mathbf{K}_{(\cdot)\boldsymbol{z}}\boldsymbol{\alpha}^*$ with

$$\boldsymbol{\alpha}^* = \underset{\boldsymbol{\alpha} \in \mathbb{R}^M}{\arg\min} \sum_{i=1}^{N} \frac{(f^{(\boldsymbol{z})}(x_i) + \varepsilon_i - \mathbf{K}_{x_i\boldsymbol{z}}\boldsymbol{\alpha})^2}{\Sigma_{ii}} + \|\boldsymbol{\alpha}\|_{\mathbf{K}_{\boldsymbol{zz}}}^2 \qquad \begin{aligned} f^{(\boldsymbol{z})}(\boldsymbol{x}) &\sim \mathrm{N}(\boldsymbol{0}, \mathbf{K}_{\boldsymbol{xz}}\mathbf{K}_{\boldsymbol{zz}}^{-1}\mathbf{K}_{\boldsymbol{zx}}) \\ \boldsymbol{\varepsilon} &\sim \mathrm{N}(\boldsymbol{0}, \boldsymbol{\Sigma}). \end{aligned} \tag{13}$$

A full derivation is given in Appendix C. Exact implementation of (13) is precluded by the need to sample from a Gaussian with covariance $\mathbf{K}_{\boldsymbol{xz}}\mathbf{K}_{\boldsymbol{zz}}^{-1}\mathbf{K}_{\boldsymbol{zx}}$. However, we identify this matrix as a Nyström approximation to $\mathbf{K}_{\boldsymbol{xx}}$. Thus, we can approximate (13) by replacing $f^{(\boldsymbol{z})}$ with $f$, which can be sampled with Fourier features (4). The approximation error is small when $M$ is large and the inducing points are close enough to the data. Our experiments in Appendix C show it to be negligible in practice. With this, we apply the stochastic estimator (7) to the inducing point sampling objective.

The inducing point objectives differ from those presented in previous sections in that there are $\mathcal{O}(M)$ and not $\mathcal{O}(N)$ learnable parameters, and we may choose the value of $M$ and locations $\boldsymbol{z}$ freely. The cost of inducing point representer weight updates is thus $\mathcal{O}(SM)$, where $S$ is the number of samples. This contrasts with the $\mathcal{O}(M^3)$ update cost of stochastic gradient variational Gaussian processes [23]. Figure 2 shows that SGD with $M \approx 100$k inducing points matches the performance of regular SGD on HOUSEELECTRIC ($N \approx 2$M), but is an order of magnitude faster.

### 3.4 The Implicit Bias and Posterior Geometry of Stochastic Gradient Descent

We have detailed an SGD-based scheme for obtaining approximate samples from a posterior Gaussian process. Despite SGD's significantly lower cost per-iteration than CG, its convergence to the true optima, shown in Figure 3, is much slower with respect to both Euclidean representer weight space, and the reproducing kernel Hilbert space (RKHS) induced by the kernel. Despite this, the predictions obtained by SGD are very close to those of the exact GP, and effectively achieve the same test RMSE. Moreover, Figure 4 shows the SGD posterior on a 1D toy task exhibits error bars of the correct width close to the data, and which revert smoothly to the prior far away from the data. Empirically, differences between the SGD and exact posteriors concentrate at the borders of data-dense regions.

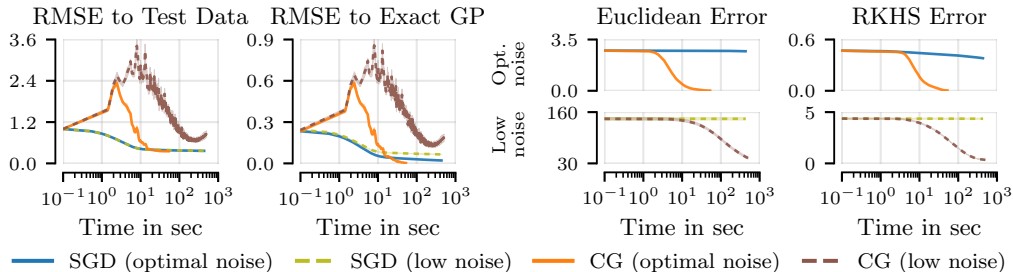

Figure 3: Convergence of GP posterior mean with SGD and CG as a function of time (on an A100 GPU) on the ELEVATORS dataset ($N \approx 16$k) while setting the noise scale to (i) maximize exact GP marginal likelihood and (ii) to $10^{-3}$, labeled *low noise*. We plot, in left-to-right order, test RMSE, RMSE to the exact GP mean at the test inputs, representer weight error $\|\boldsymbol{v} - \boldsymbol{v}^*\|_2$, and RKHS error $\|\mu_{f|\boldsymbol{y}} - \mu_{\text{SGD}}\|_{H_k}$. In the latter two plots, the low-noise setting is shown on the bottom.

We now argue the behavior seen in Figure 4 is a general feature of SGD: one can expect it to obtain good performance even in situations where it does not converge to the exact solution. Consider posterior function samples in pathwise form, namely $(f \mid \boldsymbol{y})(\cdot) = f(\cdot) + \mathbf{K}_{(\cdot)\boldsymbol{x}}\boldsymbol{v}$, where $f \sim \text{GP}(0, k)$ is a prior function sample and $\boldsymbol{v}$ are the learnable representer weights. We characterize the behavior of SGD-computed approximate posteriors by splitting the input space $X$ into 3 regions, which we call the *prior*, *interpolation*, and *extrapolation* regions. This is done as follows.

*(I) The Prior Region.* This corresponds to points sufficiently distant from the observed data. Here, for kernels that decay over space, the canonical basis functions $k(x_i, \cdot)$ go to zero. Thus, both the true posterior and any approximations formulated pathwise revert to the prior. More precisely, let $X = \mathbb{R}^d$, let $k$ satisfy $\lim_{c \to \infty} k(x', c \cdot x) = 0$ for all $x'$ and $x$ in $X$, and let $(f \mid \boldsymbol{y})(\cdot)$ be given by $(f \mid \boldsymbol{y})(\cdot) = f(\cdot) + \mathbf{K}_{(\cdot)\boldsymbol{x}}\boldsymbol{v}$, with $\boldsymbol{v} \in \mathbb{R}^N$. Then, by passing the limit through the sum, for any fixed $\boldsymbol{v}$, it follows immediately that $\lim_{c \to \infty}(f \mid \boldsymbol{y})(c \cdot x) = f(c \cdot x)$. Therefore, SGD cannot incur error in regions which are sufficiently far away from the data. This effect is depicted in Figure 4.

*(II) The Interpolation Region.* This includes points close to the training data. We characterize this region through subspaces of the RKHS, where we show SGD incurs small error.

Let $\mathbf{K}_{\boldsymbol{x}\boldsymbol{x}} = \mathbf{U}\boldsymbol{\Lambda}\mathbf{U}^T$ be the eigendecomposition of the kernel matrix. We index the eigenvalues $\boldsymbol{\Lambda} = \text{diag}(\lambda_1, \ldots, \lambda_N)$ in descending order. Define the *spectral basis functions* as eigenvector-weighed linear combinations of canonical basis functions

$$u^{(i)}(\cdot) = \sum_{j=1}^{N} \frac{U_{ji}}{\sqrt{\lambda_i}} k(x_j, \cdot). \tag{14}$$

These functions—which also appear in kernel principal component analysis—are orthonormal with respect to the RKHS inner product. To characterize them further, we lift the Courant–Fischer characterization of eigenvalues and eigenvectors to the RKHS $H_k$ induced by $k$, obtaining the expression

$$u^{(i)}(\cdot) = \operatorname*{arg\,max}_{u \in H_k} \left\{ \sum_{i=1}^{N} u(x_i)^2 : \|u\|_{H_k} = 1, \langle u, u^{(j)} \rangle_{H_k} = 0, \forall j < i \right\}. \tag{15}$$

This means in particular that the top spectral basis function, $u^{(1)}(\cdot)$, is a function of fixed RKHS norm—that is, of fixed degree of smoothness, as defined by the kernel $k$—which takes maximal values at the observations $x_1, .., x_n$. Thus, $u^{(1)}$ will be large near clusters of observations. The same will be true for the subsequent spectral basis functions, which also take maximal values at the observations, but are constrained to be RKHS-orthogonal to previous spectral basis functions. A proof of the preceding expression is given in Appendix E.3. Figure 4 confirms that the top spectral basis functions are indeed centered on the observed data.

Empirically, SGD matches the true posterior in this region. We formalize this observation by showing that SGD converges quickly in the directions spanned by spectral basis functions with large eigenvalues. Let $\text{proj}_{u^{(i)}}(\cdot)$ be the orthogonal projection onto the subspace spanned by $u^{(i)}$.

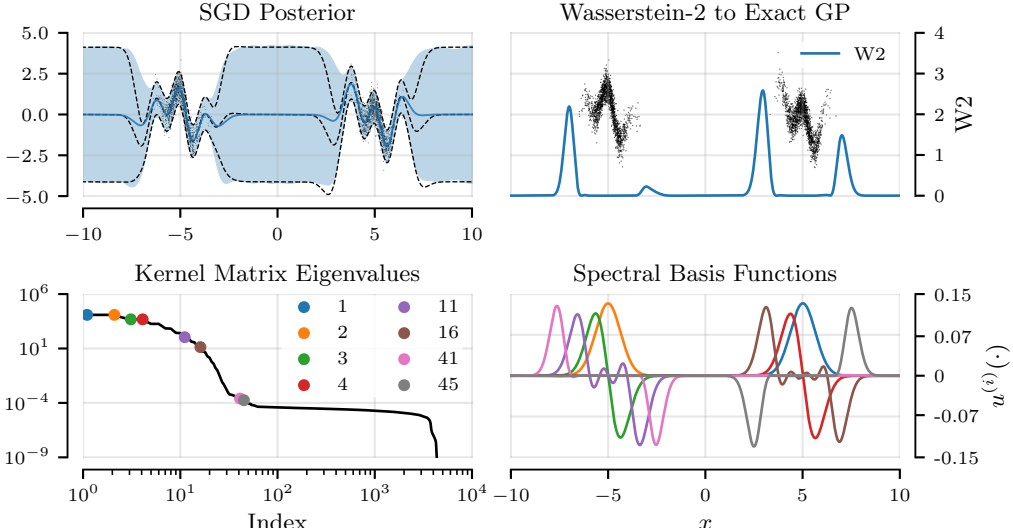

Figure 4: SGD error and spectral basis functions. Top-left: SGD (blue) and exact GP (black, dashed) fit to a $N = 10k$, $d = 1$ toy regression dataset. Top-right: 2-Wasserstein distance (W2) between both processes' marginals. The W2 values are low near the data (interpolation region) and far away from the training data. The error concentrates at the edges of the data (extrapolation region). Bottom: The low-index spectral basis functions lie on the interpolation region, where the W2 error is low, while functions of index 10 and larger lie on the extrapolation region where the error is large.

**Proposition 1.** *Let $\delta > 0$. Let $\boldsymbol{\Sigma} = \sigma^2 \mathbf{I}$ for $\sigma^2 > 0$. Let $\mu_{\text{SGD}}$ be the predictive mean obtained by Polyak-averaged SGD after $t$ steps, starting from an initial set of representer weights equal to zero, and using a sufficiently small learning rate of $0 < \eta < \frac{\sigma^2}{\lambda_1(\lambda_1 + \sigma^2)}$. Assume the stochastic estimate of the gradient is $G$-sub-Gaussian. Then, with probability $1 - \delta$, we have for $i = 1, .., N$ that*

$$\left\| \text{proj}_{u^{(i)}} \mu_{f|\boldsymbol{y}} - \text{proj}_{u^{(i)}} \mu_{\text{SGD}} \right\|_{H_k} \leq \frac{1}{\sqrt{\lambda_i}} \left( \frac{\|\boldsymbol{y}\|_2}{\eta \sigma^2 t} + G \sqrt{\frac{2}{t} \log \frac{|\mathcal{I}|}{\delta}} \right). \tag{16}$$

The proof, as well as an additional pointwise convergence bound and a variant that handles projections onto general subspaces spanned by basis functions, are provided in Appendix E.1. In general, we expect $G$ to be at most $\mathcal{O}(\lambda_1^2 \|\boldsymbol{y}\|_\infty)$ with high probability.

The result extends immediately from the posterior mean to posterior samples. As consequence, *SGD converges to the posterior GP quickly in the data-dense region*, namely where the spectral basis functions corresponding to large eigenvalues are located. Since convergence speed on the span of each basis function is independent of the magnitude of the other basis functions' eigenvalues, SGD can perform well even when the kernel matrix is ill-conditioned. This is shown in Figure 3.

*(III) The Extrapolation Region.* This can be found by elimination from the input space of the prior and interpolation regions, in both of which SGD incurs low error. Consider the spectral basis functions $u^{(i)}(\cdot)$ with small eigenvalues. By orthogonality of $u^{(1)}, .., u^{(N)}$, such functions cannot be large near the observations while retaining a prescribed norm. Their mass is therefore placed away from the observations. SGD converges slowly in this region, resulting in a large error in its solution in both a Euclidean and RKHS sense, as seen in Figure 3. Fortunately, due to the lack of data in the extrapolation region, the excess test error incurred due to SGD nonconvergence may be low, resulting in *benign nonconvergence* [52]. Similar phenomena have been observed in the inverse problems literature, where this is called *iterative regularization* [22, 27]. Figure 4 shows the Wasserstein distance to the exact GP predictions is high in this region, as when initialized at zero SGD tends to return small representer weights, thereby reverting to the prior.

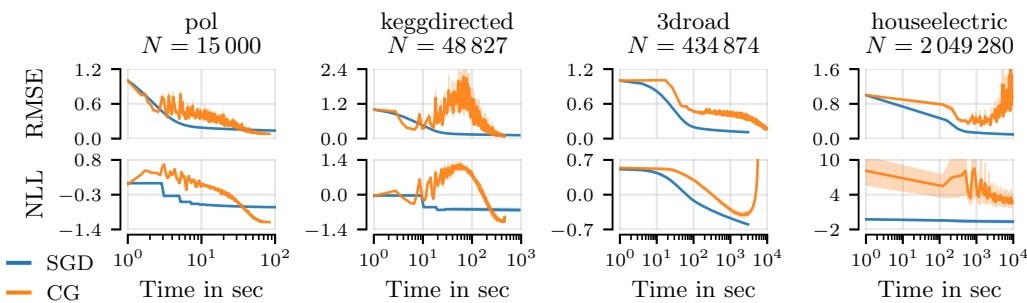

Figure 5: Test RMSE and NLL as a function of compute time on a TPUv2 core for CG and SGD.

## 4 Experiments

We now turn to empirical evaluation of SGD GPs, focusing on their predictive and decision-making properties. We compare SGD GPs with the two most popular scalable Gaussian process techniques: preconditioned conjugate gradient (CG) optimization [19, 46] and sparse stochastic variational inference (SVGP) [43, 23]. For CG, we use a pivoted Cholesky preconditioner of size 100, except in cases where this slows down convergence, where we instead report results without preconditioning. We employ the GPJax [32] SVGP implementation and use $M = 4096$ inducing points for all datasets, initializing their locations with the $K$-means algorithm. Full experimental details are in Appendix A.

### 4.1 Regression Baselines

We first compare SGD-based predictions with baselines in terms of predictive performance, scaling of computational cost with problem size, and robustness to the ill-conditioning of linear systems. Following Wang et al. [46], we consider 9 datasets from the UCI repository [16] ranging in size from $N = 15$k to $N \approx 2$M datapoints and dimensionality from $d = 3$ to $d = 90$. We report mean and standard deviation over five 90%-train 10%-test splits for the small and medium datasets, and three splits for the largest dataset. For all methods, we use a Matérn-$3/2$ kernel with a fixed set of hyperparameters obtained using maximum marginal likelihood [2], as described in Appendix A.1.

We run SGD for 100k steps, with a fixed batch size of 512 for both the mean function and samples. For CG, we run a maximum of 1000 steps for datasets with $N \leq 500$k, and a tolerance of 0.01. On the four largest datasets, CG's per-step cost is too large to run 1000 steps: instead, we run 100 steps, which takes roughly 9 hours per function sample on a TPUv2 core. For SVGP, we learn the variational parameters for $M = 4096$ inducing points by maximizing the ELBO with Adam until convergence. For all methods, we estimate predictive variances for log-likelihood computations from 64 function samples drawn using pathwise conditioning (2).

**Predictive performance and scalability with the number of inputs.** Our complete set of results is provided in Table 1, including test RMSE, test negative log-likelihood (NLL) and compute time needed to obtain the predictive mean on a single core of a TPUv2 device. Drawing multiple samples requires repeating this computation, which we perform in parallel. In the small setting ($N \leq 20$k), taking 100k steps of SGD presents a compute cost comparable to running CG to tolerance, which usually takes around 500-800 steps. Here, CG converges to the exact solution, while SGD tends to present increased test error due to non-convergence. In the large setting ($N \geq 100$k), where neither method converges within the provided compute budget, SGD achieves better RMSE and NLL and results. SVGP always converges faster than CG and SGD, but it only performs best on the BUZZ dataset, which can likely be summarised well by $M = 4096$ inducing points.

From Figure 5, we see that SGD makes the vast majority of its progress in prediction space in its first few iterations, improving roughly monotonically with the number of steps. Thus, early stopping after 100k iterations incurs only moderate errors. In contrast, CG's initial steps actually increase test error, resulting in very poor performance if stopped too early. This interacts poorly with the number of CG steps needed, and the per-step cost, which generally grow with increased amounts of data [41].

Table 1: Regression task mean and std-err for GP predictive mean RMSE, low-noise RMSE (†), TPUv2 node hours used to obtain the predictive mean, and negative-log-likelihood (NLL) computed with variances estimated from 64 function samples. SVGP is omitted for the low noise setting, where it fails to run. Metrics are reported for the datasets normalized to zero mean and unit variance.

| | Dataset
N | POL
15000 | ELEVATORS
16599 | BIKE
17379 | PROTEIN
45730 | KEGGDIR
48827 | 3DROAD
434874 | SONG
515345 | BUZZ
583250 | HOUSEELEC
2049280 |
|---|---|---|---|---|---|---|---|---|---|---|
| RMSE | SGD | $0.13 \pm 0.00$ | $0.38 \pm 0.00$ | $0.11 \pm 0.00$ | $\mathbf{0.51 \pm 0.00}$ | $0.12 \pm 0.00$ | $\mathbf{0.11 \pm 0.00}$ | $\mathbf{0.80 \pm 0.00}$ | $0.42 \pm 0.01$ | $\mathbf{0.09 \pm 0.00}$ |
| | CG | $\mathbf{0.08 \pm 0.00}$ | $\mathbf{0.35 \pm 0.00}$ | $\mathbf{0.04 \pm 0.00}$ | $0.50 \pm 0.00$ | $\mathbf{0.08 \pm 0.00}$ | $0.15 \pm 0.01$ | $0.85 \pm 0.03$ | $1.41 \pm 0.08$ | $0.87 \pm 0.14$ |
| | SVGP | $0.10 \pm 0.00$ | $0.37 \pm 0.00$ | $0.07 \pm 0.00$ | $0.57 \pm 0.00$ | $0.09 \pm 0.00$ | $0.49 \pm 0.01$ | $0.81 \pm 0.00$ | $\mathbf{0.33 \pm 0.00}$ | $0.11 \pm 0.01$ |
| RMSE † | SGD | $\mathbf{0.13 \pm 0.00}$ | $\mathbf{0.38 \pm 0.00}$ | $0.11 \pm 0.00$ | $\mathbf{0.51 \pm 0.00}$ | $\mathbf{0.12 \pm 0.00}$ | $\mathbf{0.11 \pm 0.00}$ | $\mathbf{0.80 \pm 0.00}$ | $\mathbf{0.42 \pm 0.01}$ | $\mathbf{0.09 \pm 0.00}$ |
| | CG | $0.16 \pm 0.01$ | $0.68 \pm 0.09$ | $\mathbf{0.05 \pm 0.01}$ | $3.03 \pm 0.23$ | $9.79 \pm 1.06$ | $0.34 \pm 0.02$ | $0.83 \pm 0.02$ | $5.66 \pm 1.14$ | $0.93 \pm 0.19$ |
| | SVGP | — | — | — | — | — | — | — | — | — |
| Minutes | SGD | $3.51 \pm 0.01$ | $3.51 \pm 0.01$ | $5.70 \pm 0.02$ | $\mathbf{7.10 \pm 0.01}$ | $15.2 \pm 0.02$ | $27.6 \pm 11.4$ | $220 \pm 14.5$ | $347 \pm 61.5$ | $162 \pm 54.3$ |
| | CG | $\mathbf{2.18 \pm 0.32}$ | $\mathbf{1.72 \pm 0.60}$ | $\mathbf{2.81 \pm 0.22}$ | $9.07 \pm 1.68$ | $\mathbf{12.5 \pm 1.99}$ | $85.2 \pm 36.0$ | $195 \pm 2.31$ | $351 \pm 48.3$ | $157 \pm 0.41$ |
| | SVGP | $21.2 \pm 0.27$ | $21.3 \pm 0.12$ | $20.5 \pm 0.02$ | $20.8 \pm 0.04$ | $20.8 \pm 0.05$ | $\mathbf{21.0 \pm 0.12}$ | $\mathbf{24.7 \pm 0.05}$ | $\mathbf{25.6 \pm 0.05}$ | $\mathbf{20.0 \pm 0.03}$ |
| NLL | SGD | $-0.70 \pm 0.02$ | $0.47 \pm 0.00$ | $-0.48 \pm 0.08$ | $0.64 \pm 0.01$ | $-0.62 \pm 0.07$ | $\mathbf{-0.60 \pm 0.00}$ | $1.21 \pm 0.00$ | $0.83 \pm 0.07$ | $\mathbf{-1.09 \pm 0.04}$ |
| | CG | $\mathbf{-1.17 \pm 0.01}$ | $0.38 \pm 0.00$ | $\mathbf{-2.62 \pm 0.06}$ | $0.62 \pm 0.01$ | $\mathbf{-0.92 \pm 0.10}$ | $16.27 \pm 0.45$ | $1.36 \pm 0.07$ | $2.38 \pm 0.08$ | $2.07 \pm 0.58$ |
| | SVGP | $-0.71 \pm 0.01$ | $0.43, \pm 0.00$ | $-1.27 \pm 0.02$ | $0.86 \pm 0.01$ | $-0.70 \pm 0.02$ | $0.67 \pm 0.02$ | $1.22 \pm 0.00$ | $\mathbf{0.25 \pm 0.04}$ | $-0.89 \pm 0.10$ |

**Robustness to kernel matrix ill-conditioning.** GP models are known to be sensitive to kernel matrix conditioning. We explore how this affects the algorithms under consideration by fixing the noise variance to a low value of $\sigma^2 = 10^{-6}$ and running them on our regression datasets. Table 1 shows the performance of CG severely degrades on all datasets and, for SVGP, optimization diverges for all datasets. SGD's results remain essentially-unchanged. This is because the noise only changes the smallest kernel matrix eigenvalues substantially and these do not affect convergence for the top spectral basis functions. This mirrors results presented previously for the ELEVATORS dataset in Figure 3.

**Regression with large numbers of inducing points.** We demonstrate the inducing point variant of our method, presented in Section 3.3, on HOUSEELECTRIC, our largest dataset ($N = 2$M). We select varying numbers of inducing points with a $K$-nearest-neighbor algorithm, described in Appendix A.2. Figure 2 shows the time required for 100k SGD steps scales roughly linearly with inducing points. It takes 68m for full SGD and 50m, 25m, and 17m for $M = 1099$k, 728k, and 218k, respectively. Performance in terms of RMSE and NLL degrades less than 10% even when using 218k points.

## 4.2 Large-scale Parallel Thompson Sampling

A fundamental goal of scalable Gaussian processes is to produce uncertainty estimates useful for sequential decision making. Motivated by problems in large-scale recommender systems, where both the initial dataset and the total number of users queried are simultaneously large [36, 17], we benchmark SGD on a large-scale Bayesian optimization task.

We draw a target function from a GP prior $g \sim \mathrm{GP}(0, k)$ and optimize it on $X = [0, 1]^d$ using parallel Thompson sampling [25]. That is, we choose $x_{\mathrm{new}} = \arg\max_{x \in X}(f \mid \boldsymbol{y})(\cdot)$ for a set of posterior function samples drawn in parallel. We compute these samples using pathwise conditioning with each respective scalable GP method. For each function sample maximum, we evaluate $y_{\mathrm{new}} = g(x_{\mathrm{new}}) + \varepsilon$ with $\varepsilon \sim \mathrm{N}(0, 10^{-6})$ and add the pair $(x_{\mathrm{new}}, y_{\mathrm{new}})$ to the training data. We use an acquisition batch size of 1000 samples, and maximize them with a multi-start gradient descent-based approach described in Appendix A.3. We set the search space dimensionality to $d = 8$, the largest considered by Wilson et al. [50], and initialize all methods with a dataset of 50k observations sampled uniformly at random from $X$. To eliminate model misspecification confounding, we use a Matérn-$3/2$ kernel and consider length scales of $(0.1, 0.2, 0.3, 0.4, 0.5)$ for both the target function and our models. For each length scale, we repeat the experiment for 10 seeds.

In large-scale Bayesian optimization, training and posterior function optimization costs can become significant, and predictions may be needed on demand. For this reason, we consider two variants of our experiment with different levels of compute. In the small-compute setting, SGD is run for 15k steps, SVGP is given $M = 1024$ inducing points and 20k steps to fit the variational parameters,

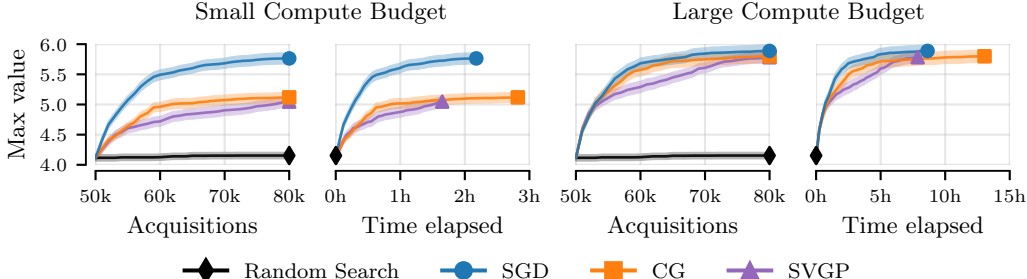

Figure 6: Maximum function values (mean and std. err.) obtained by Thompson sampling with our approximate inference methods as a function of acquisition steps and of compute time on an A100 GPU. The latter includes time spent drawing function samples and finding their maxima. All methods share a starting dataset of 50k points and we take 30 Thompson steps, acquiring 1000 points in each.

and CG is run for 10 steps. In the large-compute setting, all methods are run for 5 times as many optimization steps.

Mean results and standard errors, across length scales and seeds, are presented in Figure 6. We plot the maximum function value achieved by each method. In the small-compute setting, the ∼1.5 (A100 GPU) hours required for 30 Thompson steps with SVGP and SGD are dominated by the algorithm used to maximise the models' posterior samples. In contrast, CG takes roughly twice the time, requiring almost 3h of wall-clock time. Despite this, SGD makes the largest progress per acquisition step, finding a target function value that improves upon the initial training set maximum twice as much as the other inference methods. VI and CG perform comparably, with the latter providing slightly more progress both per acquisition step and unit of time. Despite their limited compute budget, all methods outperform random search by a significant margin. In the large-compute setting, all inference methods achieve a similar maximum target value by the end of the experiment. CG and SGD make similar progress per acquisition step but SGD is faster per unit of time. SVGP is slightly slower than CG per both. In summary, our results suggest that SGD can be an appealing uncertainty quantification technique for large-scale GP-based sequential decision making.

## 5    Conclusion

In this work, we explored using stochastic gradient algorithms to approximately compute Gaussian process posterior means and function samples. We derived optimization objectives with linear and sublinear cost for both. We showed that SGD can produce accurate predictions—even in cases when it does not converge to an optimum. We developed a spectral characterization of the effects of non-convergence, showing that it manifests itself mainly through error in an extrapolation region located away—but not too far away—from the observations. We benchmarked SGD on regression tasks of various scales, achieving state-of-the-art performance for sufficiently large or ill-conditioned settings. On a Thompson sampling benchmark, where well-calibrated uncertainty is paramount, SGD matches the performance of more expensive baselines at a fraction of the computational cost.

## Acknowledgments

We are grateful to David R. Burt for suggesting a few tricks which helped us complete the development of our theory and thank Bruno Mlodozeniec for identifying and correcting a small mistake in the proof of Appendix E.1. JAL and SP were supported by the University of Cambridge Harding Distinguished Postgraduate Scholars Programme. JA acknowledges support from Microsoft Research, through its PhD Scholarship Programme, and from the EPSRC. JMHL acknowledges support from a Turing AI Fellowship under grant EP/V023756/1. AT was supported by Cornell University, jointly via the Center for Data Science for Enterprise and Society, the College of Engineering, and the Ann S. Bowers College of Computing and Information Science. This work has been performed using resources provided by the Cambridge Tier-2 system operated by the University of Cambridge Research Computing Service (http://www.hpc.cam.ac.uk) funded by an EPSRC Tier-2 capital grant. This work was also supported with Cloud TPUs from Google's TPU Research Cloud (TRC).

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

# A Experimental Setup

We employ the CG implementation of the *JAX SciPy* module and follow Wang et al. [46] in using a pivoted Cholesky preconditioner of size 100. Our preconditioner implementation resembles the implementation of the *TensorFlow Probability* library. For a small subset of datasets, we find the preconditioner to lead to slower convergence, and we report the results for conjugate gradients without preconditioning instead.

In addition to the experiment hyperparameters described in Section 4, for all methods, we use 2000 random Fourier features to draw each prior function used for computing posterior function samples via pathwise conditioning in (4).

For SGD, at each step, we draw 100 new Fourier features to estimate the regularizer term. In all SGD experiments, we use a Nesterov momentum value of 0.9 and Polyak averaging, following Antorán et al. [3]. For all regression experiments we use a learning rate of 0.5 to estimate the mean function representer weights, and a learning rate of 0.1 to draw samples. For Thompson sampling, we use a learning rate of 0.3 for the mean and 0.0003 for the samples. In both settings, we perform gradient clipping using `optax.clip_by_global_norm` with `max_norm` set to 0.1.

Code available at: HTTPS://GITHUB.COM/CAMBRIDGE-MLG/SGD-GP.

## A.1 Hyperparameter Selection for Regression

We use a zero prior mean function and the Matérn-$3/2$ kernel, and share hyperparameters across all methods, including baselines. For each dataset, we choose a homoscedastic Gaussian noise variance, kernel variance and a separate length scale per input. For datasets with less than 50k observations, we tune these hyperparameters to maximize the exact GP marginal likelihood. The cubic cost of this procedure makes it intractable at a larger scale: instead, for datasets with more than 50k observations, we obtain hyperparameters using the following procedure:

1. From the training data, select a *centroid* data point uniformly at random.
2. Select a subset of 10k data points with the smallest Euclidean distance to the centroid.
3. Find hyperparameters by maximizing the exact GP marginal likelihood using this subset.
4. Using 10 different centroids, repeat the preceding steps and average the hyperparameters.

This approach avoids aliasing bias [1, 5] due to data subsampling and is tractable for large datasets.

## A.2 Inducing Point Selection

For the inducing point SGD experiment shown in Figure 2, we choose inducing points as a subset of the training points. Due to the large number of datapoints in our dataset, we found $k$-means to converge very slowly. Instead, we develop an ad-hoc point elimination algorithm based on $k$-nearest-neighbors (KNN). We use the KNN implementation *ANNOY*, which we first run on the HOUSEELECTRIC dataset with `num_trees` set to 50. We then iterate through our dataset, retrieving 100 nearest neighbors for each original point. Of these 100 nearest neighbors, only the ones closer to the original point than some length scale parameter, in terms of $\ell_2$ distance, are selected. If the number of points selected is larger than one, we eliminate the original point from our dataset and we also eliminate other points which are within the length scale neighborhood of the original and selected points simultaneously. The selected points are kept. We vary the number of points eliminated by our algorithm through the modification of the length scale parameter.

## A.3 Maximizing Posterior Functions for Thompson Sampling

We maximize a random acquisition function sampled from the GP posterior in a three step process:

1. Evaluate the posterior function sample at a large number of nearby input locations. We find nearby locations using a combination of exploration and exploitation. For exploration, we sample locations uniformly at random from $[0,1]^d$. For exploitation, we first subsample the training data with probabilities proportional to the observed objective function values and then add Gaussian noise $\varepsilon_{\text{nearby}} \sim \mathrm{N}(0, \sigma_{\text{nearby}}^2)$, where $\sigma_{\text{nearby}} = l/2$ and $l$ is the kernel

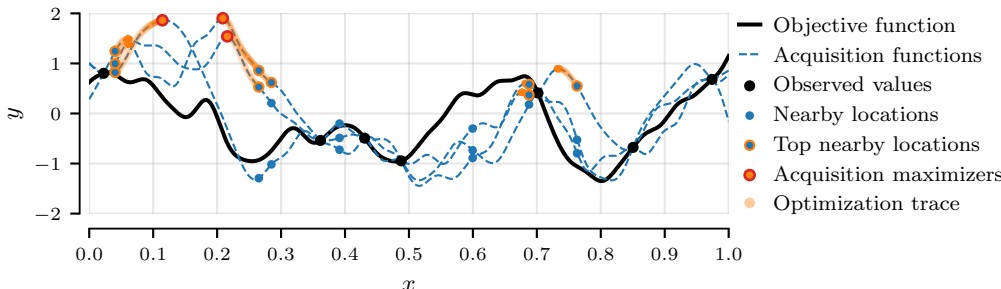

Figure 7: Illustration of a single Thompson sampling acquisition maximization step on a 1D problem.

    length scale. Throughout experiments, we find 10% of nearby locations using the uniform exploration strategy and 90% using the exploitation strategy.

2. Select the nearby locations which have the highest acquisition function values. To find the top nearby locations, we first try 50k nearby locations and then identify the location with the highest acquisition function value. We repeat this process 30 times, finding and evaluating a total of 1.5m nearby locations and obtaining a total of 30 top nearby locations.

3. Maximize the acquisition function with gradient-based optimization, using the top nearby locations as initialization. After optimization, the best location becomes $x_{\text{new}}$, the maximizer at which the true objective function will be evaluated in the next acquisition step. Initializing at the top nearby locations, we perform 100 steps of Adam on the sampled acquisition function, with a learning rate of 0.001 to find the maximizer.

In every Thompson step, we perform this three step process in parallel for 1000 random acquisition functions sampled from the GP posterior, resulting in a total of 1000 $x_{\text{new}}$, which will be added to the training data and evaluated at the objective function. Note that, although we share the initial nearby locations between sampled acquisition functions, each acquisition function will, in general, produce distinct top nearby locations and maximizers. Figure 7 illustrates a single Thompson step on a 1D problem using 3 acquisition functions, 7 nearby locations and 3 top nearby locations.

## B   Additional Experimental Results

Figure 8 provides optimization traces for SGD, CG and their low noise variants on all datasets with under 50k points. For these, we can compute the exact GP predictions, allowing us to evaluate error with respect to the exact GP in prediction space, representer weight space and in the RKHS. We observe the same trends reported in the main text. SGD decreases its prediction error monotonically, while CG does not. However, both take a similar amount of time to converge on smaller datasets and CG obtains lower error in the end. While SGD makes negligible progress in representer weight space, CG finds the optimal representer weights. CG's time to convergence is greatly increased in the low noise setting, but SGD is practically unaffected.

Table 2 provides quantitative results for inducing point SGD on the HOUSEELECTRIC dataset. SGD's time to convergence is shown to scale roughly linearly in the number of (inducing) points observed. However, for this dataset, keeping only 10% of observations and thus obtaining 10× faster

Table 2: Time to convergence and predictive performance for all methods under consideration, including inducing point SGD, on the HOUSEELECTRIC dataset.

| Model | Inducing SGD | | | Standard SGD | CG | SVGP |
|---|---|---|---|---|---|---|
| $M$ | 218 782 | 431 489 | 1 099 206 | 1 844 352 | 1 844 352 | 4 096 |
| RMSE | **0.08 ± 0.00** | **0.08 ± 0.00** | **0.08 ± 0.01** | 0.09 ± 0.00 | 0.87 ± 0.14 | 0.11 ± 0.01 |
| Minutes | **16.7 ± 0.54** | 24.6 ± 5.40 | 49.6 ± 10.2 | 162 ± 54.3 | 157 ± 0.41 | 20.0 ± 0.03 |
| NLL | **-1.10 ± 0.05** | **-1.11 ± 0.04** | **-1.13 ± 0.04** | **-1.09 ± 0.04** | 2.07 ± 0.58 | -0.89 ± 0.10 |

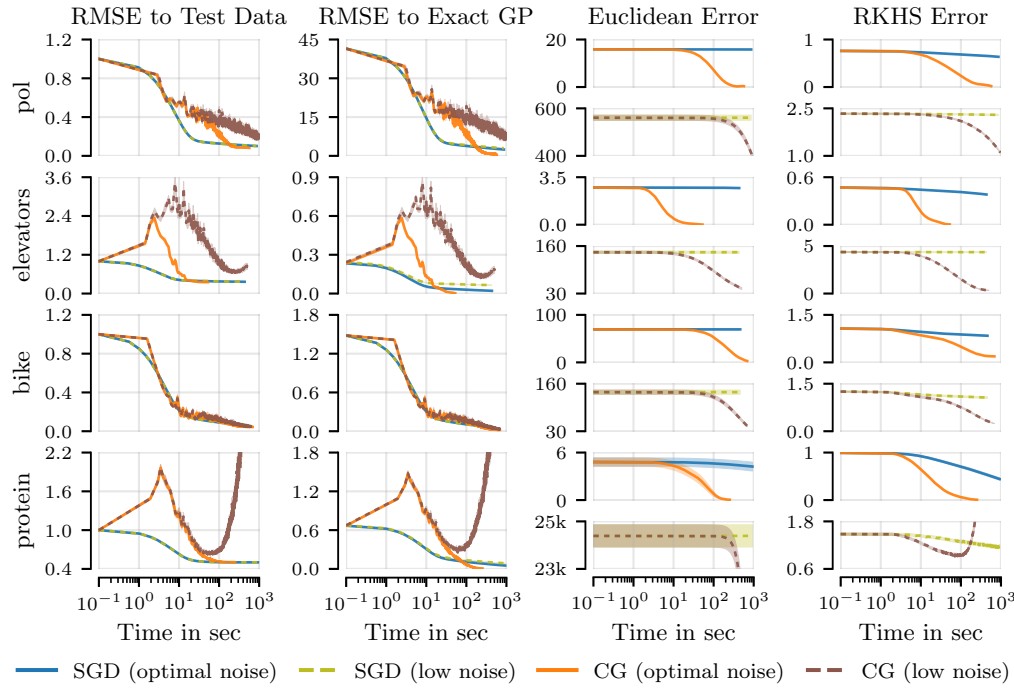

Figure 8: Convergence of GP posterior mean with SGD and CG as a function of time (on an A100 GPU) on the POL ($N \approx$ 15k), ELEVATORS ($N \approx$ 16k), BIKE ($N \approx$ 17k) and PROTEIN ($N \approx$ 46k) datasets while setting the noise scale to (i) maximize exact GP marginal likelihood and (ii) to $10^{-3}$, labeled *low noise*. We plot, in left-to-right order, test RMSE, RMSE to the exact GP mean at the test inputs, representer weight error $\|\boldsymbol{v} - \boldsymbol{v}^*\|_2$, and RKHS error $\|\mu_{f|\boldsymbol{y}} - \mu_{\mathrm{SGD}}\|_{H_k}$.

Table 3: UCI benchmark with subset of data on the smaller datasets, using a randomly-sampled subset with size that varies as a percentage of the full data.

| Dataset | | POL | ELEVATORS | BIKE | PROTEIN | KEGGDIR |
|---|---|---|---|---|---|---|
| $N$ | | 15000 | 16599 | 17379 | 45730 | 48827 |
| RMSE | 5% | $0.20 \pm 0.01$ | $0.44 \pm 0.00$ | $0.18 \pm 0.01$ | $0.71 \pm 0.01$ | $0.11 \pm 0.00$ |
| | 10% | $0.16 \pm 0.00$ | $0.41 \pm 0.01$ | $0.13 \pm 0.00$ | $0.66 \pm 0.00$ | $0.11 \pm 0.00$ |
| NLL | 5% | $-0.43 \pm 0.02$ | $0.57 \pm 0.01$ | $-0.61 \pm 0.09$ | $0.99 \pm 0.01$ | $-0.76 \pm 0.10$ |
| | 10% | $-0.66 \pm 0.02$ | $0.51 \pm 0.01$ | $-1.28 \pm 0.07$ | $0.91 \pm 0.01$ | $-0.82 \pm 0.09$ |

convergence leaves performance unaffected. This suggests the dataset can be summarized well by a small number of points. Indeed, SVGP obtains almost as strong performance as SGD in terms of RMSE with only 4096 inducing points. SVGP's NLL is weaker however, which is consistent with known issues of uncertainty overestimation when using a too small amount of inducing points. On the other hand, the large and potentially redundant nature of this dataset makes the corresponding optimization problem ill-conditioned, hurting CG's performance. Tables 3 and 4 provide additional baseline results for the *subset of data* approximation, which behaves similarly to SVGP.

Finally, Figure 9 shows all methods' Thompson sampling performance as a function of both compute time and acquisition steps for each individual true function length scale considered in our experiments. Differences among methods are more pronounced in the small compute budget setting. Here, SVGP performs well—on par with SGD—in the large length scale setting, where many observations can likely be summarized with 1024 inducing points. CG suffers from slow convergence due to ill-conditioning here. On the other hand, CG performs on par with SGD in the better-conditioned small length scale setting, while SVGP suffers. In the large compute setting, all methods perform similarly per acquisition step for all length scales except the small one, where SVGP suffers.

Table 4: UCI benchmark with subset of data on the larger datasets, using a randomly-sampled subsets of fixed size which are chosen to reflect typical GPU memory limitations. Numbers which are better than or competitive with respect to Table 1 are printed in boldface.

| Dataset | | 3DROAD | SONG | BUZZ | HOUSEELEC |
|---------|-----|--------|------|------|-----------|
| $N$ | | 434874 | 515345 | 583250 | 2049280 |
| RMSE | 25k | $0.23 \pm 0.00$ | $0.82 \pm 0.00$ | $0.33 \pm 0.00$ | $\mathbf{0.06 \pm 0.01}$ |
| | 50k | $0.16 \pm 0.00$ | $0.81 \pm 0.00$ | $\mathbf{0.32 \pm 0.00}$ | $\mathbf{0.06 \pm 0.00}$ |
| NLL | 25k | $-0.42 \pm 0.01$ | $1.22 \pm 0.00$ | $0.30 \pm 0.06$ | $-0.53 \pm 0.33$ |
| | 50k | $\mathbf{-0.78 \pm 0.01}$ | $\mathbf{1.20 \pm 0.00}$ | $\mathbf{0.26 \pm 0.06}$ | $-0.49 \pm 0.39$ |

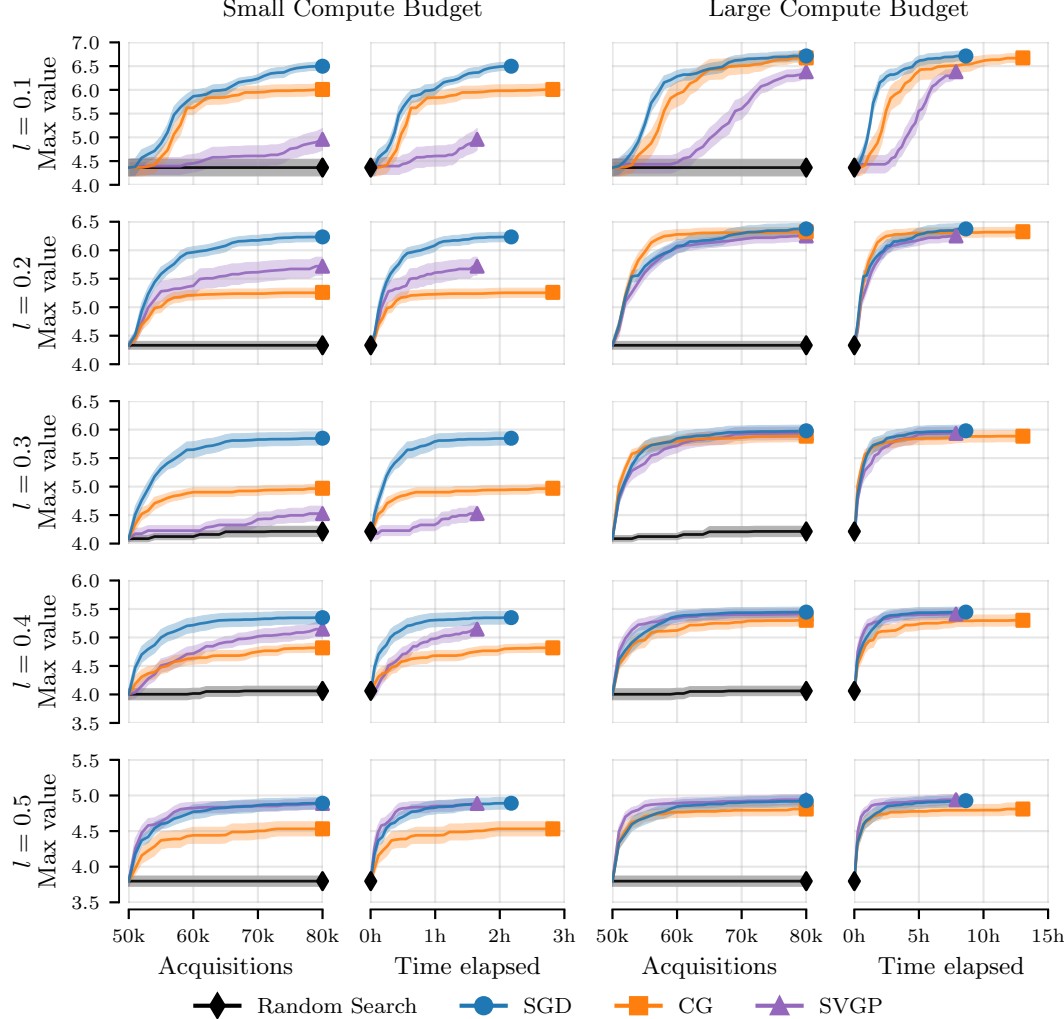

Figure 9: Maximum function values (mean and std. err.) obtained by Thompson sampling with our approximate inference methods as a function of acquisition steps and of compute time on an A100 GPU. The latter includes time spent drawing function samples and finding their maxima. All methods share a starting dataset of 50k points and we take 30 Thompson steps, acquiring 1000 points in each. Different length scales $l$ exhibit different behaviors: CG performs better in settings with smaller length scales and better conditioning, SVGP tends to perform better in settings with larger setting and increased smoothness, SGD performs well in both settings.

## C Inducing Points

Let $\boldsymbol{z} \in X^M$ be a set of $M > 0$ inducing point locations. The variational inducing point framework of Titsias [43, 44] substitutes our observed targets $\boldsymbol{y}$ with the inducing targets $\boldsymbol{u} \in \mathbb{R}^M$. Our Gaussian process, conditional on a set of inducing locations and targets, is given by the mean and covariance functions

$$\mu_{f|\boldsymbol{u}}(\cdot) = \mathbf{K}_{(\cdot)\boldsymbol{z}}\mathbf{K}_{\boldsymbol{zz}}^{-1}\boldsymbol{u} \qquad\qquad k_{f|\boldsymbol{u}}(\cdot,\cdot') = \mathbf{K}_{(\cdot,\cdot')} - \mathbf{K}_{(\cdot)\boldsymbol{z}}\mathbf{K}_{\boldsymbol{zz}}^{-1}\mathbf{K}_{\boldsymbol{z}(\cdot')}. \tag{17}$$

Titsias [43] provides a closed-form expression for the variational distribution $q = \mathrm{N}(\boldsymbol{\mu}^{(\boldsymbol{z})}, \mathbf{K}^{(\boldsymbol{z})})$ over inducing targets $\boldsymbol{u}$ which minimizes the associated Kullback–Leibler divergence, namely

$$\boldsymbol{\mu}^{(\boldsymbol{z})} = \mathbf{K}_{\boldsymbol{zz}}(\mathbf{K}_{\boldsymbol{zz}} + \mathbf{K}_{\boldsymbol{zx}}\boldsymbol{\Sigma}^{-1}\mathbf{K}_{\boldsymbol{xz}})^{-1}\mathbf{K}_{\boldsymbol{zx}}\boldsymbol{\Sigma}^{-1}\boldsymbol{y} \tag{18}$$

$$\mathbf{K}^{(\boldsymbol{z})} = \mathbf{K}_{\boldsymbol{zz}}(\mathbf{K}_{\boldsymbol{zz}} + \mathbf{K}_{\boldsymbol{zx}}\boldsymbol{\Sigma}^{-1}\mathbf{K}_{\boldsymbol{xz}})^{-1}\mathbf{K}_{\boldsymbol{zz}}. \tag{19}$$

In turn, Matthews et al. [30] shows that these expressions lift to a respective infinite-dimensional Kullback–Leibler minimization problem between the variational and true posterior Gaussian processes. Using this, we obtain a variational posterior with mean and covariance functions

$$\mu_{f|\boldsymbol{y}}^{(\boldsymbol{z})}(\cdot) = \mathbf{K}_{(\cdot)\boldsymbol{z}}(\mathbf{K}_{\boldsymbol{zz}} + \mathbf{K}_{\boldsymbol{zx}}\boldsymbol{\Sigma}^{-1}\mathbf{K}_{\boldsymbol{xz}})^{-1}\mathbf{K}_{\boldsymbol{zx}}\boldsymbol{\Sigma}^{-1}\boldsymbol{y} \tag{20}$$

$$k_{f|\boldsymbol{y}}^{(\boldsymbol{z})}(\cdot,\cdot') = \mathbf{K}_{(\cdot,\cdot')} + \mathbf{K}_{(\cdot)\boldsymbol{z}}((\mathbf{K}_{\boldsymbol{zz}} + \mathbf{K}_{\boldsymbol{zx}}\boldsymbol{\Sigma}^{-1}\mathbf{K}_{\boldsymbol{xz}})^{-1} - \mathbf{K}_{\boldsymbol{zz}}^{-1})\mathbf{K}_{\boldsymbol{z}(\cdot')}. \tag{21}$$

These expressions will be our starting point. We now write this Gaussian process posterior in a pathwise manner and derive stochastic estimators for the corresponding representer weights.

### C.1 Pathwise Representation of the Kullback–Leibler-optimal Variational Distribution

The pathwise expression of Section 3.3 can be derived by restricting the domain of the respective optimization problem which describes the exact posterior samples. This is done by replacing the reproducing kernel Hilbert space with a subset consisting of the span of a set of canonical basis functions centered at the inducing points, as described by Wild et al. [47], Theorem 5 for the posterior mean. Rather than deriving the respective result from scratch, we will verify its correctness by means of computing the relevant means and covariances. Define

$$(f^{(\boldsymbol{z})} \mid \boldsymbol{y})(\cdot) = f(\cdot) + \mathbf{K}_{(\cdot)\boldsymbol{z}}\mathbf{K}_{\boldsymbol{zz}}^{-1}\mathbf{K}_{\boldsymbol{zx}}(\mathbf{K}_{\boldsymbol{xz}}\mathbf{K}_{\boldsymbol{zz}}^{-1}\mathbf{K}_{\boldsymbol{zx}} + \boldsymbol{\Sigma})^{-1}(\boldsymbol{y} - f^{(\boldsymbol{z})}(\boldsymbol{x}) - \boldsymbol{\varepsilon})$$
$$\boldsymbol{\varepsilon} \sim \mathrm{N}(\mathbf{0}, \boldsymbol{\Sigma}) \qquad f \sim \mathrm{GP}(0, k) \qquad f^{(\boldsymbol{z})}(\cdot) = \mathbf{K}_{(\cdot)\boldsymbol{z}}\mathbf{K}_{\boldsymbol{zz}}^{-1}f(\boldsymbol{z}). \tag{22}$$

We now calculate the moments of this Gaussian process' marginal distributions and show them to match those of the KL-optimal variational Gaussian process given in (20). Write

$$\mathbb{E}[(f^{(\boldsymbol{z})} \mid \boldsymbol{y})(\cdot)] = \mathbf{K}_{(\cdot)\boldsymbol{z}}\mathbf{K}_{\boldsymbol{zz}}^{-1}\mathbf{K}_{\boldsymbol{zx}}(\mathbf{K}_{\boldsymbol{xz}}\mathbf{K}_{\boldsymbol{zz}}^{-1}\mathbf{K}_{\boldsymbol{zx}} + \boldsymbol{\Sigma})^{-1}\boldsymbol{y} \tag{23}$$

$$= \mathbf{K}_{(\cdot)\boldsymbol{z}}\mathbf{K}_{\boldsymbol{zz}}^{-1}\mathbf{K}_{\boldsymbol{zx}}\boldsymbol{\Sigma}^{-1}(\mathbf{K}_{\boldsymbol{xz}}\mathbf{K}_{\boldsymbol{zz}}^{-1}\mathbf{K}_{\boldsymbol{zx}}\boldsymbol{\Sigma}^{-1} + \mathbf{I})^{-1}\boldsymbol{y} \tag{24}$$

$$= \mathbf{K}_{(\cdot)\boldsymbol{z}}(\mathbf{K}_{\boldsymbol{zx}}\boldsymbol{\Sigma}^{-1}\mathbf{K}_{\boldsymbol{xz}} + \mathbf{K}_{\boldsymbol{zz}})^{-1}\mathbf{K}_{\boldsymbol{zx}}\boldsymbol{\Sigma}^{-1}\boldsymbol{y} \tag{25}$$

$$= \mu_{f|\boldsymbol{y}}^{(\boldsymbol{z})}(\cdot) \tag{26}$$

and

$$\mathrm{Cov}((f^{(\boldsymbol{z})} \mid \boldsymbol{y})(\cdot) - \mu_{f|\boldsymbol{y}}^{(\boldsymbol{z})}(\cdot)) \tag{27}$$

$$= \mathbb{E}((f^{(\boldsymbol{z})} \mid \boldsymbol{y})(\cdot) - \mu_{f|\boldsymbol{y}}^{(\boldsymbol{z})}(\cdot), (f^{(\boldsymbol{z})} \mid \boldsymbol{y})(\cdot') - \mu_{f|\boldsymbol{y}}^{(\boldsymbol{z})}(\cdot')) \tag{28}$$

$$= \mathbf{K}_{(\cdot,\cdot')} - \mathbf{K}_{(\cdot)\boldsymbol{z}}\mathbf{K}_{\boldsymbol{zz}}^{-1}\mathbf{K}_{\boldsymbol{zx}}(\mathbf{K}_{\boldsymbol{xz}}\mathbf{K}_{\boldsymbol{zz}}^{-1}\mathbf{K}_{\boldsymbol{zx}} + \boldsymbol{\Sigma})^{-1}\mathbf{K}_{\boldsymbol{xz}}\mathbf{K}_{\boldsymbol{zz}}^{-1}\mathbf{K}_{\boldsymbol{z}(\cdot')} \tag{29}$$

$$= \mathbf{K}_{(\cdot,\cdot')} + \mathbf{K}_{(\cdot)\boldsymbol{z}}\mathbf{K}_{\boldsymbol{zz}}^{-1}\left(-\mathbf{I} + \mathbf{I} - \mathbf{K}_{\boldsymbol{zx}}\boldsymbol{\Sigma}^{-1}(\mathbf{K}_{\boldsymbol{xz}}\mathbf{K}_{\boldsymbol{zz}}^{-1}\mathbf{K}_{\boldsymbol{zx}}\boldsymbol{\Sigma}^{-1} + \mathbf{I})^{-1}\mathbf{K}_{\boldsymbol{xz}}\mathbf{K}_{\boldsymbol{zz}}^{-1}\right)\mathbf{K}_{\boldsymbol{z}(\cdot')} \tag{30}$$

$$= \mathbf{K}_{(\cdot,\cdot')} + \mathbf{K}_{(\cdot)\boldsymbol{z}}\mathbf{K}_{\boldsymbol{zz}}^{-1}\left(-\mathbf{I} + (\mathbf{K}_{\boldsymbol{zx}}\boldsymbol{\Sigma}^{-1}\mathbf{K}_{\boldsymbol{xz}}\mathbf{K}_{\boldsymbol{zz}}^{-1} + \mathbf{I})^{-1}\right)\mathbf{K}_{\boldsymbol{z}(\cdot')} \tag{31}$$

$$= \mathbf{K}_{(\cdot,\cdot')} + \mathbf{K}_{(\cdot)\boldsymbol{z}}\left(-\mathbf{K}_{\boldsymbol{zz}}^{-1} + (\mathbf{K}_{\boldsymbol{zx}}\boldsymbol{\Sigma}^{-1}\mathbf{K}_{\boldsymbol{xz}} + \mathbf{K}_{\boldsymbol{zz}})^{-1}\right)\mathbf{K}_{\boldsymbol{z}(\cdot')} \tag{32}$$

$$= k_{f|\boldsymbol{y}}^{(\boldsymbol{z})}(\cdot,\cdot') \tag{33}$$

which recovers (21), as claimed.

## C.2 Derivation of inducing point optimization objectives

We now derive the optimization objectives we use for inducing points.

**The MAP objective** (12).  To find the optimization objective for the inducing point mean function's representer weights, we apply (23) and begin from

$$\mu_{f|\boldsymbol{y}}^{(\boldsymbol{z})}(\cdot) = \mathbf{K}_{(\cdot)\boldsymbol{z}}(\mathbf{K}_{\boldsymbol{z}\boldsymbol{x}}\boldsymbol{\Sigma}^{-1}\mathbf{K}_{\boldsymbol{x}\boldsymbol{z}} + \mathbf{K}_{\boldsymbol{z}\boldsymbol{z}})^{-1}\mathbf{K}_{\boldsymbol{z}\boldsymbol{x}}\boldsymbol{\Sigma}^{-1}\boldsymbol{y} = \mathbf{K}_{(\cdot)\boldsymbol{z}}\boldsymbol{v}^*. \tag{34}$$

Now, we recognize $(\mathbf{K}_{\boldsymbol{z}\boldsymbol{x}}\boldsymbol{\Sigma}^{-1}\mathbf{K}_{\boldsymbol{x}\boldsymbol{z}} + \mathbf{K}_{\boldsymbol{z}\boldsymbol{z}})^{-1}\mathbf{K}_{\boldsymbol{z}\boldsymbol{x}}\boldsymbol{\Sigma}^{-1}\boldsymbol{y} = \boldsymbol{v}^*$ as the expression for the optimizer of a ridge-regularized linear regression problem—see Bishop [7], Chapter 3—with parameters $\boldsymbol{v}$, features $\mathbf{K}_{\boldsymbol{x}\boldsymbol{z}}$, Gaussian noise of covariance $\boldsymbol{\Sigma}$, and regularizer $\mathbf{K}_{\boldsymbol{z}\boldsymbol{z}}$. This means that its respective optimization problem is

$$\underset{\boldsymbol{v}\in\mathbb{R}^M}{\arg\min} \quad \|\boldsymbol{y} - \mathbf{K}_{\boldsymbol{x}\boldsymbol{z}}\boldsymbol{v}\|_{\boldsymbol{\Sigma}^{-1}}^2 + \|\boldsymbol{v}\|_{\mathbf{K}_{\boldsymbol{z}\boldsymbol{z}}}^2. \tag{35}$$

**The sampling objective** (13).  For the uncertainty reduction term's representer weights with inducing points, we also leverage (23), giving

$$\mathbf{K}_{(\cdot)\boldsymbol{z}}\mathbf{K}_{\boldsymbol{z}\boldsymbol{z}}^{-1}\mathbf{K}_{\boldsymbol{z}\boldsymbol{x}}(\mathbf{K}_{\boldsymbol{x}\boldsymbol{z}}\mathbf{K}_{\boldsymbol{z}\boldsymbol{z}}^{-1}\mathbf{K}_{\boldsymbol{z}\boldsymbol{x}} + \boldsymbol{\Sigma})^{-1}(f^{(\boldsymbol{z})}(\boldsymbol{x}) + \boldsymbol{\varepsilon}) \tag{36}$$

$$= \mathbf{K}_{(\cdot)\boldsymbol{z}}(\mathbf{K}_{\boldsymbol{z}\boldsymbol{x}}\boldsymbol{\Sigma}^{-1}\mathbf{K}_{\boldsymbol{x}\boldsymbol{z}} + \mathbf{K}_{\boldsymbol{z}\boldsymbol{z}})^{-1}\mathbf{K}_{\boldsymbol{z}\boldsymbol{x}}\boldsymbol{\Sigma}^{-1}(f^{(\boldsymbol{z})}(\boldsymbol{x}) + \boldsymbol{\varepsilon}) \tag{37}$$

$$= \mathbf{K}_{(\cdot)\boldsymbol{z}}\boldsymbol{\alpha}^*. \tag{38}$$

This is again therefore the solution to a ridge-regularized quadratic problem, but now the targets are given by the random variable $(f^{(\boldsymbol{z})}(\boldsymbol{x}) + \boldsymbol{\varepsilon})$. We thus write the objective

$$\underset{\boldsymbol{\alpha}\in\mathbb{R}^M}{\arg\min} \quad \|f^{(\boldsymbol{z})}(\boldsymbol{x}) + \boldsymbol{\varepsilon} - \mathbf{K}_{\boldsymbol{x}\boldsymbol{z}}\boldsymbol{\alpha}\|_{\boldsymbol{\Sigma}^{-1}}^2 + \|\boldsymbol{\alpha}\|_{\mathbf{K}_{\boldsymbol{z}\boldsymbol{z}}}^2. \tag{39}$$

## C.3 On the Error in the Nyström Approximation $\mathbf{K}_{\boldsymbol{x}\boldsymbol{z}}\mathbf{K}_{\boldsymbol{z}\boldsymbol{z}}^{-1}\mathbf{K}_{\boldsymbol{z}\boldsymbol{x}} \approx \mathbf{K}_{\boldsymbol{x}\boldsymbol{x}}$

Figure 10 compares the KL-optimal inducing point posterior GP with that obtained when applying the approximation presented in Section 3.3. That is, taking the prior function samples which we fit with the representer weighed canonical basis functions to be $f(\boldsymbol{x})$ with $f \sim \text{GP}(0, k)$ instead of $f^{(\boldsymbol{z})}(\boldsymbol{x}) = \mathbf{K}_{\boldsymbol{x}\boldsymbol{z}}\mathbf{K}_{\boldsymbol{z}\boldsymbol{z}}^{-1}f(\boldsymbol{z})$. This amounts to approximating the Nyström-type matrix $\mathbf{K}_{\boldsymbol{x}\boldsymbol{z}}\mathbf{K}_{\boldsymbol{z}\boldsymbol{z}}^{-1}\mathbf{K}_{\boldsymbol{z}\boldsymbol{x}}$ with its exact-posterior counterpart $\mathbf{K}_{\boldsymbol{x}\boldsymbol{x}}$.  The difference between them is exactly equal to the posterior covariance: by posterior contraction, both of these matrices become very similar if there is an inducing point placed sufficiently close to every data point. In practice, this tends to occur when an inducing point is placed within roughly a half-length-scale of every observation. This is effectively what is needed for inducing point methods to provide a good approximation of the exact GP. This is reflected in Figure 10, where we see that our approximate inducing point posterior differs from the exact inducing point posterior only in situations where the latter fails to be a good approximation to the exact GP in the first place. This manifests as the approximate method providing larger error bars. When the number of inducing points increases, both methods become indistinguishable from each other and the exact GP. Fortunately, the linear cost of SGD in the number of inducing points allows us to use a very large number of these in practice.

# D Low-variance Sampling Objectives

Here, we show that the modified sampling objective (10) matches the standard sample-then-optimize objective (9) up to a constant. We then compare both objectives' minibatch gradient variance.

## D.1 Standard and Modified Sample-then-Optimize Objectives: Equivalent Gradients

Let $\mathbf{L}\mathbf{L}^T = \boldsymbol{\Sigma}$ be the Cholesky factorization of the noise covariance, let $f(\boldsymbol{x}) \sim \text{N}(\mathbf{0}, \mathbf{K}_{\boldsymbol{x}\boldsymbol{x}})$, and let $\boldsymbol{s} \sim \text{N}(\mathbf{0}, \mathbf{I})$. Consider the two objectives at hand, namely

$$\|f(\boldsymbol{x}) + \mathbf{L}\boldsymbol{s} - \mathbf{K}_{\boldsymbol{x}\boldsymbol{x}}\boldsymbol{\alpha}\|_{\boldsymbol{\Sigma}^{-1}}^2 + \|\boldsymbol{\alpha}\|_{\mathbf{K}_{\boldsymbol{x}\boldsymbol{x}}}^2 \tag{40}$$

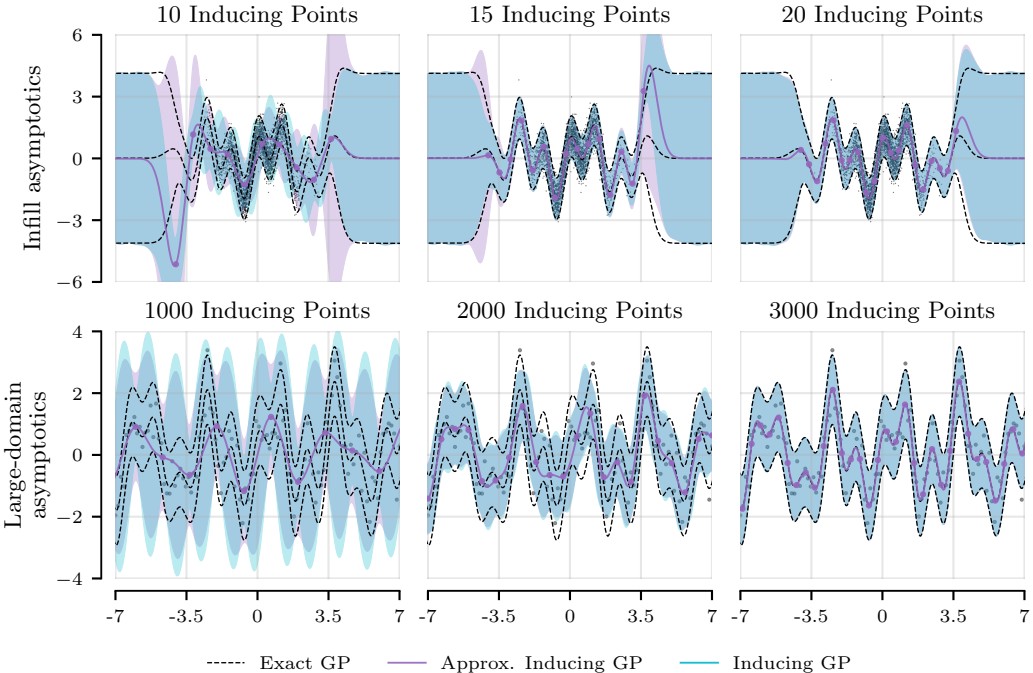

Figure 10: Comparison of exact and approximate inducing point posteriors for a GP with squared exponential kernel and 10k data points generated using the true regression function $\sin(2x) + \cos(5x)$ under two different data-generation schemes: *infill asymptotics*, which considers $x_i \sim \mathrm{N}(0,1)$, and *large-domain asymptotics*, which considers $x_i$ on an evenly spaced grid with fixed spacing. We see that the approximation needed to apply inducing points is only inaccurate in situations where the inducing point posterior itself has significant error, which generally manifests itself as error bars that are larger than those of the exact posterior.

and

$$\|f(\boldsymbol{x}) - \mathbf{K}_{\boldsymbol{xx}}\boldsymbol{\alpha}\|^2_{\boldsymbol{\Sigma}^{-1}} + \left\|\boldsymbol{\alpha} - \mathbf{L}^{-T}\boldsymbol{s}\right\|^2_{\mathbf{K}_{\boldsymbol{xx}}}. \tag{41}$$

We show these are equal up to a constant by showing they have the same derivatives. Taking derivative with respect to $\boldsymbol{\alpha}$, we have

$$\frac{\partial}{\partial\boldsymbol{\alpha}}\|f(\boldsymbol{x}) + \mathbf{L}\boldsymbol{s} - \mathbf{K}_{\boldsymbol{xx}}\boldsymbol{\alpha}\|^2_{\boldsymbol{\Sigma}^{-1}} + \|\boldsymbol{\alpha}\|^2_{\mathbf{K}_{\boldsymbol{xx}}} \tag{42}$$

$$= -2\mathbf{K}_{\boldsymbol{xx}}\boldsymbol{\Sigma}^{-1}\left(f(\boldsymbol{x}) + \mathbf{L}\boldsymbol{s} - \mathbf{K}_{\boldsymbol{xx}}\boldsymbol{\alpha}\right) + 2\mathbf{K}_{\boldsymbol{xx}}\boldsymbol{\alpha} \tag{43}$$

$$= -2\mathbf{K}_{\boldsymbol{xx}}(\boldsymbol{\Sigma}^{-1}f(\boldsymbol{x}) - \boldsymbol{\Sigma}^{-1}\mathbf{K}_{\boldsymbol{xx}}\boldsymbol{\alpha} + \mathbf{L}^{-T}\boldsymbol{s} - \boldsymbol{\alpha}), \tag{44}$$

and

$$\frac{\partial}{\partial\boldsymbol{\alpha}}\|f(\boldsymbol{x}) - \mathbf{K}_{\boldsymbol{xx}}\boldsymbol{\alpha}\|^2_{\boldsymbol{\Sigma}^{-1}} + \left\|\boldsymbol{\alpha} - \mathbf{L}^{-T}\boldsymbol{s}\right\|^2_{\mathbf{K}_{\boldsymbol{xx}}} \tag{45}$$

$$= -2\mathbf{K}_{\boldsymbol{xx}}\boldsymbol{\Sigma}^{-1}\left(f(\boldsymbol{x}) - \mathbf{K}_{\boldsymbol{xx}}\boldsymbol{\alpha}\right) + 2\mathbf{K}_{\boldsymbol{xx}}(\boldsymbol{\alpha} - \mathbf{L}^{-T}\boldsymbol{s}) \tag{46}$$

$$= -2\mathbf{K}_{\boldsymbol{xx}}(\boldsymbol{\Sigma}^{-1}f(\boldsymbol{x}) - \boldsymbol{\Sigma}^{-1}\mathbf{K}_{\boldsymbol{xx}}\boldsymbol{\alpha} + \mathbf{L}^{-T}\boldsymbol{s} - \boldsymbol{\alpha}) \tag{47}$$

respectively. These expressions match, giving the claim.

### D.2  Minibatch Gradient Variance

Following Antorán et al. [3], we consider single sample minibatch gradient estimators for the data fit term in (7), that is $D = 1$. For each of the two sampling objectives under consideration, the gradient estimators are

$$\partial L_{\text{old}} = -N\,\mathbb{E}_{i\sim U(1,\dots,N)}\,\mathbf{K}_{\boldsymbol{x},x_i}(f(x_i) + \varepsilon_i - \mathbf{K}_{x_i,\boldsymbol{x}}\boldsymbol{\alpha}) \tag{48}$$

$$\partial L_{\text{new}} = -N\,\mathbb{E}_{i\sim U(1,\dots,N)}\,\mathbf{K}_{\boldsymbol{x},x_i}(f(x_i) - \mathbf{K}_{x_i,\boldsymbol{x}}\boldsymbol{\alpha}) \tag{49}$$

for (9) and (10), respectively. Since both objectives use the same Fourier feature estimator for the gradients of the regularizer, we omit these from our analysis. The variances for single sample gradient estimators of both expressions are given by

$$\text{Cov}(\partial L_{\text{old}}) = N \, \text{Cov}(\mathbf{K}_{\boldsymbol{xx}}(f(\boldsymbol{x}) + \boldsymbol{\varepsilon} - \mathbf{K}_{\boldsymbol{x},\boldsymbol{x}}\boldsymbol{\alpha})) \tag{50}$$

$$\text{Cov}(\partial L_{\text{new}}) = N \, \text{Cov}(\mathbf{K}_{\boldsymbol{xx}}(f(\boldsymbol{x}) - \mathbf{K}_{\boldsymbol{x},\boldsymbol{x}}\boldsymbol{\alpha})), \tag{51}$$

respectively. Expanding both expressions and subtracting them we arrive at

$$\text{Cov}(\partial L_{\text{old}}) - \text{Cov}(\partial L_{\text{new}}) = N \, \text{Cov}(\mathbf{K}_{\boldsymbol{xx}}\boldsymbol{\varepsilon}) - 2N \, \text{Cov}(\mathbf{K}_{\boldsymbol{xx}}\mathbf{K}_{\boldsymbol{xx}}\boldsymbol{\alpha}, \mathbf{K}_{\boldsymbol{xx}}\boldsymbol{\varepsilon}). \tag{52}$$

Whether our proposed sampling objective presents lower variance rests on the positivity of the above matrix. To analyze this, we must give concrete values to $\boldsymbol{\alpha}$. Following Antorán et al. [3], we consider two settings, initialization and convergence.

*(I) Initialization.*   Here, $\boldsymbol{\alpha} = \mathbf{0}$ and therefore

$$\text{Cov}(\partial L_{\text{old}}) - \text{Cov}(\partial L_{\text{new}}) = N \, \text{Cov}(\mathbf{K}_{\boldsymbol{xx}}\boldsymbol{\varepsilon}) = N(\mathbf{K}_{\boldsymbol{xx}}\boldsymbol{\Sigma}\mathbf{K}_{\boldsymbol{xx}}) \tag{53}$$

is a positive definite matrix. Thus, our proposed objective has lower variance.

*(II) Convergence.*   Here, $\boldsymbol{\alpha} = (\mathbf{K}_{\boldsymbol{xx}} + \boldsymbol{\Sigma})^{-1}(f(\boldsymbol{x}) + \boldsymbol{\varepsilon})$, so we have

$$\text{Cov}(\partial L_{\text{old}}) - \text{Cov}(\partial L_{\text{new}}) = N \, \text{Cov}(\mathbf{K}_{\boldsymbol{xx}}\boldsymbol{\varepsilon}) - 2N \, \text{Cov}(\mathbf{K}_{\boldsymbol{xx}}\mathbf{K}_{\boldsymbol{xx}}\boldsymbol{\alpha}, \mathbf{K}_{\boldsymbol{xx}}\boldsymbol{\varepsilon}) \tag{54}$$

$$= N\mathbf{K}_{\boldsymbol{xx}}\boldsymbol{\Sigma}\mathbf{K}_{\boldsymbol{xx}} - 2N\mathbf{K}_{\boldsymbol{xx}}^2(\mathbf{K}_{\boldsymbol{xx}} + \boldsymbol{\Sigma})^{-1}\boldsymbol{\Sigma}\mathbf{K}_{\boldsymbol{xx}}. \tag{55}$$

Letting $\boldsymbol{\Sigma} = \sigma^2\mathbf{I}$ and $\mathbf{U}\boldsymbol{\Lambda}\mathbf{U}^T = \mathbf{K}_{\boldsymbol{xx}}$ be the eigendecomposition of the kernel matrix, we have

$$N\mathbf{K}_{\boldsymbol{xx}}\boldsymbol{\Sigma}\mathbf{K}_{\boldsymbol{xx}} - 2N\mathbf{K}_{\boldsymbol{xx}}^2(\mathbf{K}_{\boldsymbol{xx}} + \boldsymbol{\Sigma})^{-1}\boldsymbol{\Sigma}\mathbf{K}_{\boldsymbol{xx}} \tag{56}$$

$$= N\mathbf{U}\boldsymbol{\Lambda}^2(\sigma^2\mathbf{I})(\mathbf{I} - 2\boldsymbol{\Lambda}(\boldsymbol{\Lambda} + \sigma^2\mathbf{I})^{-1})\mathbf{U}^T. \tag{57}$$

Thus, at convergence, whether our proposed objective reduces variance rests on whether or not the matrix $\mathbf{I} - 2\boldsymbol{\Lambda}(\boldsymbol{\Lambda} + \sigma^2\mathbf{I})^{-1}$ is positive definite.

Following Section 3.4, in practice, the representer weights only converge along the top few principal directions. For these, the corresponding eigenvalues will likely be much larger than the noise and thus $1 - 2\frac{\lambda}{\lambda+\sigma^2} < 0$. However, in the vast majority of directions, the opposite is true: this yields lower variance to the minibatch estimator of our proposed objective, which is shown in Figure 2.

# E   The Implicit Bias of SGD GPs

This section provides our main theoretical results.

## E.1   Convergence of Stochastic Gradient Descent

Before studying what happens in Gaussian processes, we first prove a fairly standard result on the convergence of SGD for a quadratic objective appearing in kernel ridge regression, which represents the posterior mean. For simplicity, we do not analyze Nesterov momentum, nor aspects such as gradient clipping. We will take the Gaussian observation noise covariance to be a constant multiple of the identity in this subsection. Recall that a random variable $z$ is called $G$-sub-Gaussian if $\mathbb{E}(\exp(\lambda(z - \mathbb{E}(z)))) \leq \exp(\frac{1}{2}G\lambda^2)$ holds for all $\lambda$. Then, a random vector is called $G$-sub-Gaussian if its dot product with any unit vector is $G$-sub-Gaussian. One can show this condition is equivalent to having tails that are no heavier than those of a Gaussian random vector, formulated appropriately. Let $\mathbf{U}\boldsymbol{\Lambda}\mathbf{U}^T = \mathbf{K}_{\boldsymbol{xx}}$ be the eigenvalue decomposition of the kernel matrix. The eigenvectors $\boldsymbol{u}_i$ and eigenvalues $\lambda_i$ are given in descending order: $\lambda_1 \geq .. \geq \lambda_N > 0$.

**Lemma 2.** *Let $\delta > 0$ and $\boldsymbol{\Sigma} = \sigma^2\mathbf{I}$ for $\sigma^2 > 0$. Let $\eta$ be a sufficiently small learning rate of $0 < \eta < \frac{\sigma^2}{\lambda_1(\lambda_1+\sigma^2)}$. Let $\boldsymbol{\alpha}^* \in \mathbb{R}^N$ be the solution of the respective linear system, and let $\overline{\boldsymbol{\alpha}}_t$ be the Polyak-averaged SGD iterate after $t$ steps, starting from an initial condition of $\boldsymbol{\alpha}_0 = \mathbf{0}$. Assume that the stochastic estimate of the gradient is unbiased and $G$-sub-Gaussian. Then, with probability $1 - \delta$, we have for any $\mathcal{I} \subseteq \{1, .., N\}$ and all $i \in \mathcal{I}$ that*

$$|\boldsymbol{u}_i^T(\boldsymbol{\alpha}^* - \overline{\boldsymbol{\alpha}}_t)| \leq \frac{1}{\lambda_i}\left(\frac{\|\boldsymbol{y}\|_2}{\eta\sigma^2 t} + G\sqrt{\frac{2}{t}\log\frac{|\mathcal{I}|}{\delta}}\right). \tag{58}$$

*Proof.* Consider the objective

$$L(\boldsymbol{\alpha}) = \frac{1}{2\sigma^2} \sum_{i=1}^{N} (y_i - \mathbf{K}_{x_i,x}\boldsymbol{\alpha})^2 + \frac{1}{2}\boldsymbol{\alpha}^T \mathbf{K}_{xx}\boldsymbol{\alpha} \tag{59}$$

and respective gradient

$$\frac{\partial L}{\partial \boldsymbol{\alpha}}(\boldsymbol{\alpha}) = \frac{1}{\sigma^2}(\mathbf{K}_{xx}^2\boldsymbol{\alpha} - \mathbf{K}_{xx}\boldsymbol{y}) + \mathbf{K}_{xx}\boldsymbol{\alpha} = \frac{1}{\sigma^2}(\mathbf{K}_{xx}(\mathbf{K}_{xx} + \sigma^2\mathbf{I})\boldsymbol{\alpha} - \mathbf{K}_{xx}\boldsymbol{y}). \tag{60}$$

Let us first look at non-stochastic gradient optimization of $L$, without Polyak averaging. The iterate $\boldsymbol{\alpha}_t$ is given by

$$\boldsymbol{\alpha}_t = \boldsymbol{\alpha}_{t-1} - \eta\frac{\partial L}{\partial \boldsymbol{\alpha}}(\boldsymbol{\alpha}_{t-1}) = \left(\mathbf{I} - \frac{\eta}{\sigma^2}\mathbf{K}_{xx}(\mathbf{K}_{xx} + \sigma^2\mathbf{I})\right)\boldsymbol{\alpha}_{t-1} + \frac{\eta}{\sigma^2}\mathbf{K}_{xx}\boldsymbol{y}. \tag{61}$$

Writing $\mathbf{M} = \left(\mathbf{I} - \frac{\eta}{\sigma^2}\mathbf{K}_{xx}(\mathbf{K}_{xx} + \sigma^2\mathbf{I})\right)$, and recalling that $\boldsymbol{\alpha}_0 = \mathbf{0}$, we thus have that

$$\boldsymbol{\alpha}_t = \frac{\eta}{\sigma^2}\sum_{j=0}^{t-1}\mathbf{M}^j\mathbf{K}_{xx}\boldsymbol{y} = \frac{\eta}{\sigma^2}(\mathbf{I} - \mathbf{M})^{-1}(\mathbf{I} - \mathbf{M}^t)\mathbf{K}_{xx}\boldsymbol{y} = \boldsymbol{\alpha}^* - (\mathbf{K}_{xx} + \sigma^2\mathbf{I})^{-1}\mathbf{M}^t\boldsymbol{y} \tag{62}$$

where, for our choice of learning rate, (i) by direct calculation using simultaneous diagonalizability of $\mathbf{M}$ and $\mathbf{K}_{xx}$, the matrix $\mathbf{M}$ has eigenvalues whose absolute values are strictly less than 1, (ii) thus, the geometric series converges, (iii) and $\mathbf{I} - \mathbf{M}$ is positive definite. Examining the error in direction $\boldsymbol{u}_i$, we use simultaneous diagonalizability to see that

$$|\boldsymbol{u}_i^T(\boldsymbol{\alpha}^* - \boldsymbol{\alpha}_t)| = |\boldsymbol{u}_i^T(\mathbf{K}_{xx} + \sigma^2\mathbf{I})^{-1}\mathbf{M}^t\boldsymbol{y}| = \frac{\left(1 - \frac{\eta\lambda_i}{\sigma^2}(\lambda_i + \sigma^2)\right)^t}{\lambda_i + \sigma^2}|\boldsymbol{u}_i^T\boldsymbol{y}| \tag{63}$$

$$\leq \frac{\left(1 - \frac{\eta\lambda_i}{\sigma^2}(\lambda_i + \sigma^2)\right)^t}{\lambda_i + \sigma^2}\|\boldsymbol{y}\|_2 \tag{64}$$

which applies to ordinary gradient descent without stochastic gradients or Polyak averaging. Next, we consider stochastic gradient optimization. We will first consider the case where the gradient is independently perturbed by $\boldsymbol{\zeta}_t \sim \mathrm{N}(\mathbf{0}, \mathbf{I})$ for each step $t > 0$, and then relax this to sub-Gaussian noise in the sequel. Specifically, we consider

$$\boldsymbol{\alpha}'_t = \boldsymbol{\alpha}'_{t-1} - \eta\left(\frac{\partial L}{\partial \boldsymbol{\alpha}}(\boldsymbol{\alpha}'_{t-1}) + \boldsymbol{\zeta}_t\right) = \boldsymbol{\alpha}_t - \eta\sum_{j=0}^{t-1}\mathbf{M}^j\boldsymbol{\zeta}_{t-j} \tag{65}$$

with $\boldsymbol{\alpha}'_0 = \boldsymbol{\alpha}_0 = \mathbf{0}$. For these iterates, consider the respective Polyak-averaged iterates denoted by $\overline{\boldsymbol{\alpha}}_t = \frac{1}{t}\sum_{j=1}^{t}\boldsymbol{\alpha}'_j$. We have

$$|\boldsymbol{u}_i^T(\boldsymbol{\alpha}^* - \overline{\boldsymbol{\alpha}}_t)| = \left|\frac{1}{t}\sum_{j=1}^{t}\boldsymbol{u}_i^T(\boldsymbol{\alpha}^* - \boldsymbol{\alpha}'_j)\right| \leq \underbrace{\left|\frac{1}{t}\sum_{j=1}^{t}\boldsymbol{u}_i^T(\boldsymbol{\alpha}^* - \boldsymbol{\alpha}_j)\right|}_{A_{i,t}} + \underbrace{\left|\frac{1}{t}\sum_{j=1}^{t}\boldsymbol{u}_i^T(\boldsymbol{\alpha}_j - \boldsymbol{\alpha}'_j)\right|}_{B_{i,t}} \tag{66}$$

by expanding the Polyak averages and applying the triangle inequality, and where we have introduced the notation $A_{i,t}$ and $B_{i,t}$. Using (63), the triangle inequality and another geometric series, we can bound the first sum as

$$|A_{i,t}| = \frac{\|\boldsymbol{y}\|_2}{(\lambda_i + \sigma^2)t}\sum_{j=1}^{t}\left(1 - \frac{\eta\lambda_i}{\sigma^2}(\lambda_i + \sigma^2)\right)^j \leq \frac{\sigma^2\|\boldsymbol{y}\|_2}{\eta\lambda_i(\lambda_i + \sigma^2)^2 t} \tag{67}$$

For the second sum, note that we can re-index the order of summation to to count each $\boldsymbol{\zeta}_j$ only once, rather than once for each Polyak average. This gives

$$B_{i,t} = \frac{\eta}{t}\sum_{j=1}^{t}\sum_{q=0}^{j-1}\boldsymbol{u}_i^T\mathbf{M}^q\boldsymbol{\zeta}_{j-q} = \frac{\eta}{t}\sum_{j=1}^{t}\left(\sum_{q=0}^{t-j}\boldsymbol{u}_i^T\mathbf{M}^q\right)\boldsymbol{\zeta}_j \tag{68}$$

$$= \frac{\eta}{t}\sum_{j=1}^{t}\sum_{q=0}^{t-j}\left(1 - \frac{\eta\lambda_i}{\sigma^2}(\lambda_i + \sigma^2)\right)^q\boldsymbol{u}_i^T\boldsymbol{\zeta}_j \tag{69}$$

where $\boldsymbol{u}_i^T \boldsymbol{\zeta}_j \sim \mathrm{N}(0,1)$ are IID. The variance of $B_i$ is bounded using another geometric series

$$\mathrm{Var}(B_{i,t}) = \mathrm{Var}\left(\frac{\eta}{t}\sum_{j=1}^{t}\sum_{q=0}^{t-j}\left(1 - \frac{\eta\lambda_i}{\sigma^2}(\lambda_i + \sigma^2)\right)^q \boldsymbol{u}_i^T \boldsymbol{\zeta}_j\right) \tag{70}$$

$$= \frac{\eta^2}{t^2}\sum_{j=1}^{t}\mathrm{Var}\left(\sum_{q=0}^{t-j}\left(1 - \frac{\eta\lambda_i}{\sigma^2}(\lambda_i + \sigma^2)\right)^q \boldsymbol{u}_i^T \boldsymbol{\zeta}_j\right) \tag{71}$$

$$= \frac{\eta^2}{t^2}\sum_{j=1}^{t}\left[\left(\sum_{q=0}^{t-j}\left(1 - \frac{\eta\lambda_i}{\sigma^2}(\lambda_i + \sigma^2)\right)^q\right)^2 \underbrace{\mathrm{Var}\left(\boldsymbol{u}_i^T \boldsymbol{\zeta}_j\right)}_{=1}\right] \tag{72}$$

$$\leq \frac{\eta^2}{t^2}\sum_{j=1}^{t}\left(\frac{\sigma^2}{\eta\lambda_i(\lambda_i + \sigma^2)}\right)^2 = \frac{\sigma^4}{\lambda_i^2(\lambda_i + \sigma^2)^2 t}. \tag{73}$$

Thus, from standard tail inequalities for Gaussian random variables, for any fixed $\delta' > 0$ we have

$$\mathbb{P}\left(|B_{i,t}| \leq \sqrt{\frac{2\sigma^4}{\lambda_i^2(\lambda_i + \sigma^2)^2 t}\log\frac{1}{\delta'}}\right) \geq 1 - \delta'. \tag{74}$$

We then take $\delta' = \delta/|\mathcal{I}|$, and apply the union bound for all indices in $\mathcal{I}$. Finally, by comparing moment generating functions, we can relax the Gaussian assumption to $G$-sub-Gaussian random variables. To complete the claim, we combine the bounds for the two sums by writing

$$|\boldsymbol{u}_i^T(\boldsymbol{\alpha}^* - \overline{\boldsymbol{\alpha}}_t)| \leq |A_{i,t}| + |B_{i,t}| \leq \frac{\sigma^2\|\boldsymbol{y}\|_2}{\eta\lambda_i(\lambda_i + \sigma^2)^2 t} + G\sqrt{\frac{2\sigma^4}{\lambda_i^2(\lambda_i + \sigma^2)^2 t}\log\frac{|\mathcal{I}|}{\delta}} \tag{75}$$

$$= \frac{\sigma^2}{\lambda_i(\lambda_i + \sigma^2)}\left(\frac{\|\boldsymbol{y}\|_2}{\eta(\lambda_i + \sigma^2)t} + G\sqrt{\frac{2}{t}\log\frac{|\mathcal{I}|}{\delta}}\right) \tag{76}$$

$$\leq \frac{1}{\lambda_i}\left(\frac{\|\boldsymbol{y}\|_2}{\eta\sigma^2 t} + G\sqrt{\frac{2}{t}\log\frac{|\mathcal{I}|}{\delta}}\right) \tag{77}$$

which gives the claim. $\qquad\square$

### E.2 Convergence of SGD in RKHS Subspaces

The idea is to use the preceding result to show that SGD converges fast with respect to a certain seminorm. Let $k$ be the kernel and let $H_k$ be its associated reproducing kernel Hilbert space. We now define the spectral basis functions from Section 3.4.

**Definition 3.** *Let $\mathbf{K}_{\boldsymbol{x}\boldsymbol{x}} = \mathbf{U}\boldsymbol{\Lambda}\mathbf{U}^T$ be the respective eigendecomposition. Define the* SPECTRAL BASIS FUNCTIONS

$$u^{(i)}(\cdot) = \sum_{j=1}^{N}\frac{U_{ji}}{\sqrt{\lambda_i}}k(x_j, \cdot). \tag{78}$$

These are precisely the basis functions which arise in kernel principal component analysis. We also introduce two subspaces of the RKHS: the span of the representer weights, and the span of a subset of spectral basis functions. These are defined as follows.

**Definition 4.** *Define the* REPRESENTER WEIGHT SPACE

$$R_{k,\boldsymbol{x}} = \mathrm{span}\{k(x_i, \cdot) : i = 1, .., N\} \subseteq H_k \tag{79}$$

*equipped with the subspace inner product.*

**Definition 5.** *Let $\mathcal{I} \subseteq \{1, .., N\}$ be an arbitrary set of indices. Define the* INTERPOLATION SUBSPACE

$$R_{k,\boldsymbol{x}}^{\mathcal{I}} = \mathrm{span}\{u^{(i)} : i \in \mathcal{I}\} \subseteq R_{k,\boldsymbol{x}}. \tag{80}$$

We can use these subspaces to define a seminorm on the RKHS, which we will use to measure convergence rates of SGD.

**Definition 6.** *Define the* INTERPOLATION SEMINORM

$$|f|_{R_{k,\boldsymbol{x}}^{\mathcal{I}}} = \|\mathrm{proj}_{R_{k,\boldsymbol{x}}^{\mathcal{I}}} f\|_{H_k}. \tag{81}$$

Here, $\mathrm{proj}_{\mathcal{S}} f$ denotes orthogonal projection of a function $f$ onto the subspace $\mathcal{S}$ of a Hilbert space: to ease notation, in cases where we project onto the span of a single vector, we write the vector in place of $\mathcal{S}$. The first order of business is to understand the relationship between the spectral basis functions and representer weight space.

**Lemma 7.** *The functions $u^{(i)}$, for $i = 1, .., N$, form an orthonormal basis of $R_{k,\boldsymbol{x}}$*

*Proof.* Let $i \neq j$. Using the reproducing property, definition of an eigenvector, the assumption that $\boldsymbol{u}$ is an eigenvector of $\mathbf{K}_{\boldsymbol{xx}}$, and orthonormality of eigenvectors, we have

$$\langle u^{(i)}, u^{(j)}\rangle_{H_k} = \frac{\boldsymbol{u}_i^T \mathbf{K}_{\boldsymbol{xx}} \boldsymbol{u}_j}{\sqrt{\lambda_i \lambda_j}} = \frac{\lambda_i}{\sqrt{\lambda_i \lambda_j}} \boldsymbol{u}_i^T \boldsymbol{u}_j = 0. \tag{82}$$

The fact that $u^{(i)}$ form a basis follows from the fact that they are linearly independent, that there are $N$ of them in total, and that by definition this equals the dimension of $R_{k,\boldsymbol{x}}$. Finally, we calculate the norm:

$$\|u^{(i)}\|_{H_k}^2 = \frac{\boldsymbol{u}_i \mathbf{K}_{\boldsymbol{xx}} \boldsymbol{u}_i^T}{\lambda_i} = 1. \tag{83}$$

$\square$

Next, we compute a change-of-basis formula between the canonical basis functions and spectral basis functions.

**Lemma 8.** *For some vector $\boldsymbol{\theta} \in \mathbb{R}^N$, let*

$$\theta(\cdot) = \sum_{i=1}^N \theta_i k(x_i, \cdot) = \sum_{j=1}^N w_j u^{(j)}(\cdot). \tag{84}$$

*Then we have*

$$\boldsymbol{w} = \boldsymbol{\Lambda}^{1/2} \mathbf{U}^T \boldsymbol{\theta} \qquad\qquad \|\theta\|_{H_k}^2 = \boldsymbol{w}^T \boldsymbol{w}. \tag{85}$$

*Proof.* By expanding $u^{(j)}$, we have

$$\theta(\cdot) = \sum_{j=1}^N w_j u^{(j)}(\cdot) = \sum_{i=1}^N \sum_{j=1}^N w_j \frac{U_{ij}}{\sqrt{\lambda_j}} k(x_i, \cdot) \tag{86}$$

which since $k(x_i, \cdot)$ is a basis means $\theta_i = \sum_{j=1}^N w_j \frac{U_{ij}}{\sqrt{\lambda_j}}$, or in matrix-vector notation $\boldsymbol{\theta} = \mathbf{U}\boldsymbol{\Lambda}^{-1/2}\boldsymbol{w}$. The final claim about norms follows from orthonormality of $u^{(j)}$ by writing

$$\|\theta\|_{H_k}^2 = \langle \theta, \theta \rangle_{H_k} = \sum_{i=1}^N \sum_{j=1}^N w_i w_j \langle u^{(i)}, u^{(j)}\rangle_{H_k} = \sum_{i=1}^N w_i^2 = \boldsymbol{w}^T \boldsymbol{w}. \tag{87}$$

$\square$

This allows us to get an explicit expression for the previously-introduced seminorm.

**Lemma 9.** *For $\theta \in R_{k,\boldsymbol{x}}$, letting $\boldsymbol{\Lambda}_{\mathcal{I}} = \mathrm{diag}(\lambda_1 \mathbb{1}_{1 \in \mathcal{I}}, .., \lambda_i \mathbb{1}_{i \in \mathcal{I}}, .., \lambda_N \mathbb{1}_{N \in \mathcal{I}})$, we have*

$$|\theta|_{R_{k,\boldsymbol{x}}^{\mathcal{I}}}^2 = \boldsymbol{\theta}^T \mathbf{U} \boldsymbol{\Lambda}_{\mathcal{I}} \mathbf{U}^T \boldsymbol{\theta} \tag{88}$$

*Proof.* Decompose $\theta$ in the above orthonormal basis, giving

$$\theta(\cdot) = \sum_{j=1}^{N} w_j u^{(j)}(\cdot). \tag{89}$$

By orthonormality, the definition of $R_{k,\boldsymbol{x}}^{\mathcal{I}}$, and properties of projections, we have

$$(\text{proj}_{R_{k,\boldsymbol{x}}^{\mathcal{I}}} \theta)(\cdot) = \sum_{j \in \mathcal{I}} w_j u^{(j)}(\cdot). \tag{90}$$

Therefore, again using orthonormality, we obtain

$$|\theta|_{R_{k,\boldsymbol{x}}^{\mathcal{I}}}^2 = \sum_{j \in \mathcal{I}} w_j^2 = \boldsymbol{\theta}^T \mathbf{U} \mathbf{\Lambda}_{\mathcal{I}} \mathbf{U}^T \boldsymbol{\theta} \tag{91}$$

which gives the claim. $\square$

With this at hand, we claim that our variant of SGD converges quickly with respect to this seminorm.

**Proposition 10.** *Under the assumptions of Lemma 2, for any $\mathcal{I}$ we have*

$$\left|\mu_{f|\boldsymbol{y}} - \mu_{\text{SGD}}\right|_{R_{k,\boldsymbol{x}}^{\mathcal{I}}} \leq \left(\frac{\|\boldsymbol{y}\|_2}{\eta\sigma^2 t} + G\sqrt{\frac{2}{t}\log\frac{|\mathcal{I}|}{\delta}}\right)\sqrt{\sum_{i \in \mathcal{I}}\frac{1}{\lambda_i}}. \tag{92}$$

*Proof.* Using Lemma 9 and linearity, we have

$$\left|\mu_{f|\boldsymbol{y}} - \mu_{\text{SGD}}\right|_{R_{k,\boldsymbol{x}}^{\mathcal{I}}} = \left|\sum_{i=1}^{N}(v_i - v_i^*)k(x_i, \cdot)\right|_{R_{k,\boldsymbol{x}}^{\mathcal{I}}} = \sqrt{(\boldsymbol{v} - \boldsymbol{v}^*)^T \mathbf{U}\mathbf{\Lambda}_{\mathcal{I}}\mathbf{U}^T(\boldsymbol{v} - \boldsymbol{v}^*)} \tag{93}$$

$$= \sqrt{\sum_{i \in \mathcal{I}}\left|\boldsymbol{u}_i^T(\boldsymbol{v} - \boldsymbol{v}^*)\right|^2 \lambda_i} \leq \sqrt{\sum_{i \in \mathcal{I}}\left|\frac{1}{\lambda_i}\left(\frac{\|\boldsymbol{y}\|_2}{\eta\sigma^2 t} + G\sqrt{\frac{2}{t}\log\frac{|\mathcal{I}|}{\delta}}\right)\right|^2 \lambda_i} \tag{94}$$

$$= \left(\frac{\|\boldsymbol{y}\|_2}{\eta\sigma^2 t} + G\sqrt{\frac{2}{t}\log\frac{|\mathcal{I}|}{\delta}}\right)\sqrt{\sum_{i \in \mathcal{I}}\frac{1}{\lambda_i}} \tag{95}$$

where the final inequality comes from substituting in the result of Lemma 2, yielding the claim. $\square$

Our main claim follows directly from the established framework.

**Proposition 1.** *Let $\delta > 0$. Let $\mathbf{\Sigma} = \sigma^2 \mathbf{I}$ for $\sigma^2 > 0$. Let $\mu_{\text{SGD}}$ be the predictive mean obtained by Polyak-averaged SGD after $t$ steps, starting from an initial set of representer weights equal to zero, and using a sufficiently small learning rate of $0 < \eta < \frac{\sigma^2}{\lambda_1(\lambda_1 + \sigma^2)}$. Assume the stochastic estimate of the gradient is $G$-sub-Gaussian. Then, with probability $1 - \delta$, we have for $i = 1, .., N$ that*

$$\left\|\text{proj}_{u^{(i)}}\mu_{f|\boldsymbol{y}} - \text{proj}_{u^{(i)}}\mu_{\text{SGD}}\right\|_{H_k} \leq \frac{1}{\sqrt{\lambda_i}}\left(\frac{\|\boldsymbol{y}\|_2}{\eta\sigma^2 t} + G\sqrt{\frac{2}{t}\log\frac{|\mathcal{I}|}{\delta}}\right). \tag{16}$$

*Proof.* Choose $\mathcal{I}$ to be a singleton, and apply Proposition 10, where by replacing $|\mathcal{I}|$ with $N$ in the final bound the claim can be made to hold with probability $1 - \delta$ for all $i = 1, .., N$ simultaneously. $\square$

This concludes the first part of the argument for why SGD is a good idea: it converges fast with respect to this seminorm. In particular, taking $\mathcal{I}$ to be the full index set, a position-dependent pointwise convergence bound follows.

**Corollary 11.** *Under the assumptions of Lemma 2, we have*

$$|\mu_{\text{SGD}}(\cdot) - \mu_{f|\boldsymbol{y}}(\cdot)| \leq \left(\frac{\|\boldsymbol{y}\|_2}{\eta\sigma^2 t} + G\sqrt{\frac{2}{t}\log\frac{|\mathcal{I}|}{\delta}}\right)\sum_{i=1}^{N}\frac{1}{\sqrt{\lambda_i}}|u^{(i)}(\cdot)|. \tag{96}$$

*Proof.* Using Lemma 8, write

$$\mu_{\mathrm{SGD}}(\cdot) - \mu_{f|\boldsymbol{y}}(\cdot) = \sum_{j=1}^{N}(v_j - v_j^*)k(x_i, \cdot) = \sum_{i=1}^{N}(w_i - w_i^*)u^{(i)}(\cdot) \tag{97}$$

where $\boldsymbol{w} - \boldsymbol{w}^* = \boldsymbol{\Lambda}^{1/2}\mathbf{U}^T(\boldsymbol{v} - \boldsymbol{v}^*)$. Applying Lemma 2 with $\mathcal{I} = \{1, .., N\}$, we conclude

$$|\mu_{\mathrm{SGD}}(\cdot) - \mu_{f|\boldsymbol{y}}(\cdot)| \leq \sum_{i=1}^{N}|w_i - w_i^*||u^{(i)}(\cdot)| = \sum_{i=1}^{N}\sqrt{\lambda_i}|\boldsymbol{u}_i^T(\boldsymbol{v} - \boldsymbol{v}^*)||u^{(i)}(\cdot)| \tag{98}$$

$$\leq \sum_{i=1}^{N}\sqrt{\lambda_i}\left|\frac{1}{\lambda_i}\left(\frac{\|\boldsymbol{y}\|_2}{\eta\sigma^2 t} + G\sqrt{\frac{2}{t}\log\frac{|\mathcal{I}|}{\delta}}\right)\right||u^{(i)}(\cdot)| \tag{99}$$

$$= \left(\frac{\|\boldsymbol{y}\|_2}{\eta\sigma^2 t} + G\sqrt{\frac{2}{t}\log\frac{|\mathcal{I}|}{\delta}}\right)\sum_{i=1}^{N}\frac{1}{\sqrt{\lambda_i}}|u^{(i)}(\cdot)|. \tag{100}$$

The claim follows. $\qquad\square$

Examining the expression, the approximation error will be high at locations $x \in X$ where $u^{(i)}(x)$ corresponding to tail eigenfunctions is large. We proceed to analyze when this occurs.

### E.3 Courant–Fischer in the Reproducing Kernel Hilbert Space

Since the preceding seminorm is induced by a projection within the RKHS, the next step is to understand what kind of functions the corresponding subspace contains. To get a qualitative view, the basic idea is to lift the Courant–Fischer characterization of eigenvalues and eigenvectors to the RKHS.

We will need the following variant of the Min-Max Theorem, which is useful for describing multiple eigenvectors at once—as solutions to a sequence of Raleigh-quotient-type optimization problems, where the next optimization problem is performed on a subspace which ensures its solution is orthogonal to all previous solutions. Mirroring the rest of this work, we consider eigenvalues in decreasing order, namely $\lambda_1 \geq .. \geq \lambda_N \geq 0$.

**Result 12.** *Let $\mathbf{A}$ be a positive semi-definite matrix. Then*

$$\lambda_i = \max_{\substack{\boldsymbol{u}\in\mathbb{R}^N\backslash\{\boldsymbol{0}\} \\ \boldsymbol{u}^T\boldsymbol{u}_j=0,\forall j<i}} \frac{\boldsymbol{u}^T\mathbf{A}\boldsymbol{u}}{\boldsymbol{u}^T\boldsymbol{u}} \tag{101}$$

*where the eigenvectors $\boldsymbol{u}_i$ are the respective maximizers.*

*Proof.* Horn and Johnson [26], Theorem 4.2.2, modified appropriately to handle eigenvalues in decreasing order, and where we choose $i_1, .., i_k = 1, .., N - i + 1$. $\qquad\square$

We will prove the main claim below in two stages: first, on representer weight space $R_{k,\boldsymbol{x}}$, then on the full RKHS $H_k$. We begin with the former.

**Lemma 13.** *We have*

$$u^{(i)}(\cdot) = \arg\max_{u\in R_{k,\boldsymbol{x}}}\left\{\sum_{i=1}^{N}u(x_i)^2 : \|u\|_{H_k} = 1, \langle u, u^{(j)}\rangle_{H_k} = 0, \forall j < i\right\}. \tag{102}$$

*Proof.* Define $\boldsymbol{w}_j = \mathbf{U}\boldsymbol{\Lambda}^{-1/2}\boldsymbol{u}_j$, where we recall that $\boldsymbol{u}_j$ are the respective eigenvectors of the kernel matrix. From the Result 12 form of the Courant–Fischer Min-Max Theorem, the reproducing property, and the explicit form of the RKHS inner product on $R_{k,\boldsymbol{x}}$ in the basis defined by the canonical basis functions, we can conclude that

$$\lambda_i = \max_{\substack{\boldsymbol{u}\in\mathbb{R}^N\backslash\{\boldsymbol{0}\} \\ \boldsymbol{u}^T\boldsymbol{u}_j=0,\forall j<i}} \frac{\boldsymbol{u}^T\mathbf{K}_{\boldsymbol{x}\boldsymbol{x}}\boldsymbol{u}}{\boldsymbol{u}^T\boldsymbol{u}} = \max_{\substack{\boldsymbol{w}\in\mathbb{R}^N\backslash\{\boldsymbol{0}\} \\ \boldsymbol{w}^T\mathbf{K}_{\boldsymbol{x}\boldsymbol{x}}\boldsymbol{w}_j=0,\forall j<i}} \frac{\boldsymbol{w}^T\mathbf{K}_{\boldsymbol{x}\boldsymbol{x}}^2\boldsymbol{w}}{\boldsymbol{w}^T\mathbf{K}_{\boldsymbol{x}\boldsymbol{x}}\boldsymbol{w}} \tag{103}$$

$$= \max_{\substack{\boldsymbol{w}\in\mathbb{R}^N\setminus\{\boldsymbol{0}\} \\ \boldsymbol{w}^T\mathbf{K}_{\boldsymbol{xx}}\boldsymbol{w}_j=0,\forall j<i}} \frac{\|\mathbf{K}_{\boldsymbol{xx}}\boldsymbol{w}\|_2^2}{\boldsymbol{w}^T\mathbf{K}_{\boldsymbol{xx}}\boldsymbol{w}} = \max_{\substack{\boldsymbol{w}\in\mathbb{R}^N\setminus\{\boldsymbol{0}\} \\ \boldsymbol{w}^T\mathbf{K}_{\boldsymbol{xx}}\boldsymbol{w}_j=0,\forall j<i}} \frac{\left\|\sum_{i=1}^N w_i k(x_i,\boldsymbol{x})\right\|_2^2}{\left\|\sum_{i=1}^N w_i k(x_i,\cdot)\right\|_{H_k}^2} \tag{104}$$

$$= \max_{\substack{u\in R_{k,\boldsymbol{x}}\setminus\{0\} \\ \langle u,u^{(j)}\rangle_{H_k}=0,\forall j<i}} \frac{\|u(\boldsymbol{x})\|_2^2}{\|u\|_{H_k}^2} = \max_{\substack{u\in R_{k,\boldsymbol{x}} \\ \|u\|_{H_k}=1 \\ \langle u,u^{(j)}\rangle_{H_k}=0,\forall j<i}} \sum_{i=1}^N u(x_i)^2 \tag{105}$$

which gives the claim. $\qquad\square$

Next, we use a Representer-Theorem-type orthogonality argument to extend the optimization problem to all of $H_k$. The argument will rely heavily on the Projection Theorem for Hilbert spaces: for an overview of the setting, see Lang [28], Chapter 5.

**Proposition 14.** *We have*

$$u^{(i)}(\cdot) = \arg\max_{u\in H_k}\left\{\sum_{i=1}^N u(x_i)^2 : \|u\|_{H_k}=1, \langle u,u^{(j)}\rangle_{H_k}=0,\forall j<i\right\}. \tag{106}$$

*Proof.* Let $H_k^{(i)}$ be the orthogonal complement of the finite-dimensional subspace $\mathrm{span}\{u^{(j)}, j=1,..,i-1\}$ in $H_k$ and note by feasibility that $u^{(i)} \in H_k^{(i)}$. Consider the decomposition of $u^{(i)}$ into its projection onto an orthogonal complement with respect to the finite-dimensional subspace $H_k^{(i)} \cap R_{k,\boldsymbol{x}}$, namely

$$u^{(i)}(\cdot) = u^{\|}(\cdot) + u^{\perp}(\cdot) \tag{107}$$

where this notation is defined as $u^{\|} = \mathrm{proj}_{H_k^{(i)}\cap R_{k,\boldsymbol{x}}} u^{(i)}$ and $u^{\perp} = u^{(i)} - u^{\|}$. Observe that

$$u^{\perp}(x_i) = \langle u^{\perp}, k(x_i,\cdot)\rangle_{H_k} = \langle \mathrm{proj}_{H_k^{(i)}} u^{\perp}, k(x_i,\cdot)\rangle_{H_k} = \langle u^{\perp}, \mathrm{proj}_{H_k^{(i)}} k(x_i,\cdot)\rangle_{H_k} = 0 \tag{108}$$

where the first equality follows from the reproducing property, the second equality follows because $u^{\perp} \in H_k^{(i)}$, the third equality follows because the projection is orthogonal and therefore self-adjoint, and last equality follows since $u^{\perp}$ by definition lives in the orthogonal complement of $H_k^{(i)} \cap R_{k,\boldsymbol{x}}$ within $H_k^{(i)}$, with the subspace inner product. Moreover, by the Projection Theorem and the fact that $H_k^{(i)}$ inherits its norm from $H_k$, note that

$$\|u^{(i)}\|_{H_k}^2 = \|u^{\|}\|_{H_k}^2 + \|u^{\perp}\|_{H_k}^2. \tag{109}$$

We now argue by contradiction: suppose that $u^{\perp} \neq 0$. This means that $\|u^{\perp}\|_{H_k} > 0$. From the above, we have

$$\sum_{i=1}^N u^{(i)}(x_i)^2 = \sum_{i=1}^N u^{\|}(x_i)^2 \qquad\qquad \|u^{\|}\|_{H_k} < \|u^{(i)}\|_{H_k}. \tag{110}$$

Define the function

$$u'(\cdot) = \frac{\|u^{(i)}\|_{H_k}}{\|u^{\|}\|_{H_k}} u^{\|}(\cdot) \tag{111}$$

and note that $\|u'\| = 1$, and $\langle u', u^{(j)}\rangle = 0$ for all $j < i$ since $u^{\|} \in H_k^{(i)}$, which makes $u'$ a feasible solution to the optimization problem considered. At the same time, it satisfies

$$\sum_{i=1}^N u'(x_i)^2 = \frac{\|u^{(i)}\|_{H_k}^2}{\|u^{\|}\|_{H_k}^2}\sum_{i=1}^N u^{\|}(x_i)^2 > \sum_{i=1}^N u^{\|}(x_i)^2 = \sum_{i=1}^N u^{(i)}(x_i)^2 \tag{112}$$

which shows that $u^{(i)}$, contrary to its definition, is not actually optimal—contradiction. We conclude that $u^{\perp} = 0$, which means that $u^{(i)} \in H_k^{(i)} \cap R_{k,\boldsymbol{x}}$, and the claim follows from Lemma 13. $\qquad\square$

This means that the top eigenvalues can be understood as the largest possible squared sums of canonical basis functions evaluated at the data, subject to RKHS-norm constraints. In situations where $k(x,\cdot)$ is continuous, non-negative, and decays as one moves away from the data, it is intuitively clear that such sums will be maximized by placing weight on canonical basis functions which cover as much of the data as possible. This mirrors the RKHS view of kernel principal component analysis.

