# OpenReview forum: "Sampling from Gaussian Process Posteriors using Stochastic Gradient Descent"
_NeurIPS.cc/2023/Conference — NeurIPS 2023 oral_

### Official Review · Reviewer_xja3 · 2023-06-20

**Soundness:** 4 excellent
**Presentation:** 3 good
**Contribution:** 3 good
**Rating:** 8
**Confidence:** 4

**Summary:**

I thank the authors for supplying the additional experiments and making the comparisons. It appears the technique is a novel and competitive method that can perform very well even for difficult datasets in the large-scale regime.

Below is the initial review, unchanged.
-------------------------------------------------------------------

The authors propose to sample from the posterior of a GP via optimisation using SGD. The authors rephrase sampling as a quadratic optimisation problem that allows an efficient approximation of the gradient using random kernel features. Then, SGD is used to find the solution. Theoretical derivations show tht the proposed algorithm gives good variance estimates in densely sampled areas as well as areas far outside the sampled regions. Further, the algorithm performs better than CG and SVGP in certain settings.

**Strengths:**

The article provides a reltive complete package with an understandable derivation, good theoretical results and experimental evaluations.

The observations of the good performance of SGD in densely sampled regions is very interesting.

**Weaknesses:**

The article omits subset of data (SoD) methods completely, both in the related work and the experimental section. [1] introduces them as as category, [2] establish them empirically as competitive method in sampling and [3] investigates the size of the subsampling dataset.

[1] A Unifying View of Sparse Approximate Gaussian Process Regression, Qui~nonero-Candela and Rasmussen (2005)

[2] A Framework for Evaluating Approximation Methods for Gaussian Process Regression, Chalupka and Murray and Williams (2013)

[3] Adaptive Cholesky Gaussian Processes, Bartels et al. (2023)

Subsampling methods perform similarly to the proposed method as they work well in densely sampled regions and also well in regions far away from the data distributions with only region of large error in the low sampled tails. This makes comparison mandatory.

The compairosn with SVGP is unfair due to the small number of inducing points (1024) and a bad optimisation algorithm for them (ADAM). It seems like the authors did not tune the competing methods well, making the resulting baselines weak.

Theoretically, the authors omit in their complexity class evaluations the number of SGD steps, which might become very large.
Further, the authors omit the standard work on GP, while refering to standard notation introduced by it:

[4] Gaussian Processes for Machine Learning, Carl Edward Rasmussen and Christopher K. I. Williams (2006)

**Questions:**

Question: Can you provide plots including a parameter study of SVGP including a stronger optimizer?

Suggestion: I would suggest to introduce SoP method in the related work, but also compare them to the proposed method in the experimental section. I would suggest to compare them empirically to the proposed method by given them a dataset size with computation budget the same as the proposed method to allow a fair comparison on wallclock time.

**Limitations:**

-

---

> ### Author Rebuttal · Authors · 2023-08-08
>
> Thank you very much for your review! Below we address the key points:
>
> **Comparison with Subset of Data Methods**
>
> Thank you for bringing SoD methods to our attention. *We ran SoD on our regression datasets.* We randomly select subsets from the training data and build exact GP models with these points. We use the complete datasets for data normalization. We provide mean results and std. err. across dataset splits and subset seeds.
>
> We consider 5% and 10% subsets for the small and medium datasets. For the larger datasets, we use 25k and 50k (largest possible on an 80GB A100 GPU) subsets. These represent roughly 5% and 10% of 3droad song and buzz, and 1.25 and 2.5% of houselectric.
>
> Due to the character limit, we do not reproduce the numbers from our paper here but we bold best results across this table and the paper.
>
> **SoD Small & Medium Datasets**
> |      |  SoD |        pol       |    elevators    |       bike      |      protein     | keggdir   |
> |:----:|:----:|:----------------:|:---------------:|:---------------:|:----------------:|-----------|
> |      |      |     N = 15000    |    N = 16599    |    N = 17379    |     N = 45730    | N = 48827 |
> | RMSE |   5%  | 0.20 ± 0.01  | 0.44 ± 0.00 | 0.18 ± 0.01  | 0.71 ± 0.01 | 0.11 ± 0.00  |
> |      |  10%  | 0.16 ± 0.00  | 0.41 ± 0.01 | 0.13 ± 0.00  | 0.66 ± 0.00 | 0.10 ± 0.00  |
> |  NLL |   5%  | -0.43 ± 0.02 | 0.57 ± 0.01 | -0.61 ± 0.09 | 0.99 ± 0.01 | -0.76 ± 0.10 |
> |      |  10%  | -0.66 ± 0.02 | 0.51 ± 0.01 | -1.28 ± 0.07 | 0.91 ± 0.01 | -0.82 ± 0.09 |
>
> **SoD Large Datasets**
> |      |  SoD |      3droad      |       song      |       buzz      |     houseelec    |
> |:----:|:----:|:----------------:|:---------------:|:---------------:|:----------------:|
> |      |      |    N = 434874    |    N = 515345   |    N = 583250   |    N = 2049280   |
> | RMSE |  25k | 0.23 ± 0.00  | 0.82 ± 0.00 | 0.33 ± 0.00 | 0.06 ± 0.01  |
> |      |  50k | 0.16 ± 0.00  | 0.81 ± 0.00 | **0.32 ± 0.00** | **0.06 ± 0.00**  |
> |  NLL |  25k | -0.42 ± 0.01 | 1.22 ± 0.00 | 0.30 ± 0.06 | -0.53 ± 0.33 |
> |      |  50k | **-0.78 ± 0.01** | **1.20 ± 0.00** | **0.26 ± 0.06** | -0.49 ± 0.39 |
>
> In terms of mean prediction, SoD does not perform best on any small or medium datasets. We also tried higher percentages: the results are similar. On the large datasets, the 50k subset performs best on buzz, where SVGP previously was the best method, and on NLL for 3droad and Buzz. Note that an exact GP on 50k points requires 40 GB of GPU memory; it is not accessible to most practitioners.
>
> In terms of error bar geometry, SoD performance is heavily dataset dependent: it performs well when the observations are largely redundant. **We graphically illustrate this with Figure 1(r) in the rebuttal PDF attached to the summary post**.
>
> To conclude, we would like to emphasize that our contribution is presenting a **novel approach for scaling up Gaussian processes**, quite dissimilar from existing ones. This is valuable due to its opportunity to unlock new avenues for GP research, not because our technique outperforms all baselines across all tasks, which it does not - and, *this is not one of our claims*, as Table 1 (in paper) clearly shows instances where SGD is outperformed by CG and SVGP.
>
>
> **SVGP Hyperparameters and optimization algorithm (ADAM)**
>
> * **Hyperparameters.** To facilitate comparisons with prior work, **our paper used the same SVGP and CG  hyperparameters as Wang et. al. 2019** (a very well-cited NeurIPS2019 paper), which studies conjugate-gradient-based Gaussian processes, and whose techniques are now used in GPyTorch.
>
> We have also trained SVGP with $M=4096$ inducing points on our 4 largest datasets (we will include the rest in the camera ready), which is rather expensive because 10k optimization steps are needed for convergence and each step's cost is cubic in the number of inducing points $\mathcal{O}(M^3)$.
>
> |   SVGP 4096   |      3droad     |       song      |       buzz      |     houseelec    |
> |:----:|:---------------:|:---------------:|:---------------:|:----------------:|
> | RMSE | 0.49 ± 0.01 | 0.81 ± 0.00 | **0.33 ± 0.00** |  0.11 ± 0.01 |
> |  NLL | 0.67 ± 0.02 | 1.22 ± 0.00 | **0.25 ± 0.04** | -0.90 ± 0.10 |
>
> Although increasing the inducing points improves SVGP's performance on all large datasets, SVGP 4096 only performs best on buzz, where the 1024 point version was already best.
>
> * **Adam.** Our understanding is that ADAM is currently the best optimizer for SVGP. It is the default in GPyTorch, GPFlow, and GPJax. We suspect the reviewer may be thinking of *full-batch* methods such as LBFGS, which **cannot be used with stochastic approximation (i.e. minibatching)**. These optimizers are SoTA for full-batch inducing point approaches such as Titsias 2009, but are incompatible with SVGP (namely, *stochastic* variational GP) which is mini-batch based.
>
> **SGD steps in the complexity analysis**
>
> This is a good question! Our work's key finding is that *SGD does not need to be run to convergence to produce good empirical performance*. On this basis, it makes sense to view the number of SGD steps as a hyperparameter. This is reflected in our paper's language; **We make no claims about the complexity of SGD Inference, only about the cost of a single optimization step** - e.g. in line 106: “(7)[our unbiased estimator] presents O(N) complexity, in contrast with the O(N^2) complexity of one CG step.”
>
> **Empirically, the number of steps needed to obtain a given performance level is, roughly independent of the dataset size**. We use 100k SGD steps across all experiments, except the Bayesian optimization experiment where we vary this parameter as part of the experiment. We find not only that this is sufficient to obtain reasonable results in all cases but also that the convergence plots (Figures 3, 5, 8) present a similar shape for numbers of observations running from 10k to 2M.
>
> **We have added citation of Rasmussen and Williams (2006)**

---

### Official Review · Reviewer_w7c5 · 2023-07-04

**Soundness:** 4 excellent
**Presentation:** 4 excellent
**Contribution:** 4 excellent
**Rating:** 9
**Confidence:** 4

**Summary:**

The paper presents a novel approach for sampling from GP posteriors based on SGD, which bypasses the need to solve the typical linear system of equations that is prevalent in both the exact variant (cubic in the number of query points) or the pathwise approximation (cubic in the number of data points). The SGD approximation is provided for both exact GPs and for inducing point approximations. Moreover, the approximation quality is investigated for three regions of varying data density, and explanations for the (occasionally poor) approximation quality are given.

**Strengths:**

- __Novel, interesting idea:__ The use of SGD for atypical objectives is interesting, and the analysis of the approach is detailed from both a theoretical and empirical perspective.

- __Clearly addressed limitations:__ The SGD approach is _not_ a silver bullet, and the authors make this clear by highlighting the approximation quality in data-dense regions (good), faraway regions (good), and interpolation regions (not as good).

- __Informative, well-designed figures:__ The various figures are not only visually appealing, but provide . Figure 1 and 4 in particular highlight the strengths and shortcomings of the method nicely, and seamlessly add intuition as to why that is.

- __Diverse Experiments:__ Experiments from both large-scale GP regression and BO are included, which demonstrates that the method is applicable and potent in both domains.

- __Clear writing:__ The paper is consistently well-formulated, pedagogical and as far as I could tell, correct. Moreover, I beieve that there has been substantial effort to provide the reader with additional intuition for why the approach is effective.

**Weaknesses:**

I struggle to find weaknesses with the paper, but remain unconvinced on its potential impact due to the relatively small niche (sampling for GPs in large data regime) that is addressed. However, I am not overly confident in this assessment, and invite the authors to challenge my opinion on this topic. For example, do the authors see opportunities for impactful follow-up work which spans other areas of ML?

**Questions:**

- __Comparison to Pathwise Sampling:__ Does the proposed method hold any advantages to pathwise sampling in a low-to-moderate data regime in terms of accuracy or complexity? At which point (in #data points) does SGD start becoming beneficial?
- __Computing the predicitve uncertainty:__ Perhaps a trivial question, but how is the predicitve uncertainty computed when one can only access the posterior mean and samples from the posterior (and not the posterior variance) in Section 4.1?

**Limitations:**

Limitations are adequately addressed.

---

> ### Author Rebuttal · Authors · 2023-08-08
>
> We would like to thank you their time in reading our work! We are thrilled that you found our writing and plots “Clear" and "Informative” and agreed that our paper "Clearly addressed limitations". We now discuss weaknesses and respond to your questions:
>
>
> **Significance:**
> > "I ... remain unconvinced on its potential impact due to the relatively small niche (sampling for GPs in large data regime) that is addressed."
>
> * **Our main motivating setting is large-scale Bayesian optimization**. Sampling from large-data GPs with fixed hyperparameters is a key component in large-scale Bayesian optimization, particularly in industrial settings. Bayesian optimization (whether under this name, or that of GP bandit algorithms) is a strong approach for the optimization of black-box systems. In particular, sampling from GPs is a core element of the Thompson sampling algorithm. Historically, due to the cost of fitting GPs, Bayesian optimization was limited to small—perhaps even toy—systems; however, work undertaken over the last 5-10 years on scaling inference in GPs, which our method is a part of, now allows for its use on an industrial scale (e.g., optimizing stock levels and recommendations at Amazon). The Thompson-sampling-based approach to Bayesian optimization is particularly well suited to parallelization and asynchronous processing, and thus combines well with such large-scale systems. We believe our approach to sampling to be particularly well-suited for Thompson sampling and easy to use due to its robustness to ill-conditioning. Thus, it has the potential to be adopted by industry users at the usual big-name tech companies (which all provide online recommendations to users and tackle other similar bandit problems, well-suited for Bayesian optimization).
>
> * More generally, there is growing interest in applying GPs to **spatiotemporal modeling** (Howes et al., PLOS Global Public Health 2023, "Spatio-temporal estimates of HIV risk group proportions for adolescent girls and young women across 13 priority countries in sub-Saharan Africa"), applications in the **physical and natural sciences** (Goḿez-Bombarelli et al., ACS Central Science 2018, "Automatic Chemical Design Using a Data-Driven Continuous Representation of Molecules") and **climate modeling** (Thompson et al., Environmental Data Science 2022, "A dependent multimodel approach to climate prediction with Gaussian processes)". Here datasets tend to be large, and as shown by Foster et al. (2009, JMLR, "Stable and Efficient Gaussian Process Calculations"), and Terenin et al. (2023, arXiv:2210.07893, "Numerically Stable Sparse Gaussian Processes via Minimum Separation using Cover Trees"), **ill-conditioned systems appear in almost all moderate-to-large-scale GP problems**. Regularization through for instance inducing point choice helps, but at the cost of bias and performance. Our work suggests an orthogonal way to handle instability is to design algorithms that tolerate ill-conditioning well.
>
>
> **Questions:**
>
> 1. **Comparison to Pathwise Sampling:**
> > "Does the proposed method hold any advantages to pathwise sampling in a low-to-moderate data regime in terms of accuracy or complexity? At which point (in #data points) does SGD start becoming beneficial?"
>
> * This is a great question! Since our approach is an approximation to efficient sampling (namely pathwise conditioning with no approximations except for the prior term) we expect it to perform worse whenever solving the involved linear systems exactly is tractable, for instance in the low-data, well-conditioned regime. From our experiments (Table 1), we estimate that the transition where SGD may start to become better than CG occurs in the 50k-100k datapoint range. Where in that range depends on kernel-matrix conditioning. For very poorly conditioned kernel matrices, SGD can perform better with only a couple of tens of thousands of points.
>
> 2. **Computing the predictive uncertainty:**
> > "How is the predictive uncertainty computed when one can only access the posterior mean and samples from the posterior (and not the posterior variance) in Section 4.1?"
>
> * For each test-point, we estimate the scalar predictive variance from 64 0-mean posterior samples $f_i(x)$ as $\frac{1}{64}\sum_{i=1}^{64} f_i(x)^2$. We do this for all methods under consideration. This is tucked away in the second paragraph of Section 4.1 - thank you for pointing this out, we will look into making this point easier to find.

---

> > ### Comment · Reviewer_w7c5 · 2023-08-10
> > **Thank you!**
> >
> > Thanks to the authors for addressing my questions. I am very much aware of the BO implications, but I value the references to other use-cases, which points to the potentially substantial impact of the work.
> >
> > This paper was a pleasure to read. Once again, I greatly appreciated the clearly addressed limitations, which should not go unnoted.
> >
> > I have increased my score to a 9.

---

### Official Review · Reviewer_Hc1f · 2023-07-05

**Soundness:** 4 excellent
**Presentation:** 4 excellent
**Contribution:** 3 good
**Rating:** 7
**Confidence:** 5

**Summary:**

This work proposes SGD GP, a method based on stochastic gradient descent to efficiently compute the GP posterior samples given fixed hyperparameters. The method relies on the pathwise conditioning GP posterior formulation and random Fourier features (RFF) approximation. The key idea is to express the GP posterior quantities as solutions to quadratic optimization problems whose objective is a sum over data points and hence SGD can be applied.

The paper shows that SGD GP produce accurate predictions. However, SGD GP can converge slowly or converge to sub-optimum. However, non-convergence behaviors of SGD only occur in region closed to the data boundary. SGD GP performs comparably to SVGP and conjugate gradients (CG) in most settings, and can outperform them in large-scale systems or ill-conditioned problems.

**Strengths:**

- The paper is overall well-written and easy to follow.

- The proposed method is novel and sound. It is shown to provide better predictive performance given the same inference time compared to SVGP and CG. What I found is most compelling is that SGD-GP seems to be a stronger alternative in large-scale or ill-conditioned systems.

- I also find the spectral analysis of SGD convergence of three different regions to be very insightful.

**Weaknesses:**

- The major weakness is that the setting the paper considers is quite limited, which is the posterior inference given fixed learned hyperparameter. I would appreciate if the authors can elaborate on the significance of the setting and why the proposed methodology is in particular important. For example, how often would the ill-conditioned systems arise in practice and how common is the case that hyperparameters are known in advance. It also seems like the method is applied to a isotropic Gaussian likelihood (see the question section below).

- The fact that SGD converges slowly in extrapolation region is a bit concerning. Especially in applications like Bayesian optimization, this is the region of high interests for exploration. Or in settings where there are distribution shifts, the under-calibrated uncertainty in this region can be an issue. In general, I found the "benign non-convergence" argument in the extrapolation region not convincing. Would appreciate if the authors can elaborate on this issue, and if if there is any potential way to alleviate the non-convergence property?

- The SGD convergence analysis is insightful. But I would appreciate a more formal mathematical characterization of three regions.

**Questions:**

- Related to the first point above, can the method be adapted for broader settings where the hyperparameters need to be learned, or the likelihood is not Gaussian? If not, what is the limiting factor there?

- Regarding the experiment evaluation, sec 4.1, is the predictive RMSE and NLL good metrics for evaluation? If I understand it correctly, SGD and CG are based on the same set of learned hyperpararmeters (and SVGP is based on a separately learned variational model). The core goal of the comparison is whether SGD recovers more faithful inference approximation compared to CG as a baseline. So I thought the valuable metric should be a distance against the "exact" inference result (e.g. CG with max. number of iterations and with high numeric precision). So I am not sure how to interpret Table 1.

- In Fig 3 and Fig 5, do you have a sense why CG errors first go up and then go down? In Fig 5 last panel (houseelectric) would you expect CG error to match SGD performance eventually if running for more time? In both figures, I think it would be helpful to provide an exact baseline (e.g. CG ran to reach tolerance like1e-3).

Minor comments.

- A few notations and figures are not very clear. E.g. Fig 4. top left panel, indicate the size of error bands (blue shaded area) and dotted black line; top right corner, indicate the black dots (observations). In Proposition 1, define G-sub-gaussian.

**Limitations:**

The limitation discussion is missing. I don't think the paper would have negative societal impact. My main concern on technical limitaiton is weakness point 1.

---

> ### Author Rebuttal · Authors · 2023-08-08
>
> We wanted to start by thanking you for your time in reading our work and providing very helpful comments! We are thrilled you found our work “well-written and easy to follow” and agreed that our results are interesting because they provide a “stronger alternative in large-scale or ill-conditioned systems”, which as we will argue below include most systems of sufficient scale.
>
> ----
>
> Below we address the weaknesses and questions:
>
> 1.  **Significance of setting**
>
>
> * *Significance*: You are correct that our proposed approach can not be used to tune hyperparameters and requires a Gaussian (*but not necessarily isotropic*) likelihood. In our view, presenting a new, scalable, way to perform posterior inference in GPs is a valuable contribution on its own which can pave the path for future work on hyperparameter selection and non-conjugate inference.
>
>
> * *The fixed hyperparameter* setting occurs in large-scale Bayesian optimization, particularly in industrial settings. Here, hyperparameters are often selected using historical (offline) data, and no updates are made online thereafter. Combining online hyperparameter updates and closed-loop systems is very challenging in practice and rigorous theory of these updates is scarce.
>
>
> * *Ill conditioned systems* appear in almost all all moderate-to-large-scale GP problems. See Foster et al. (2009, JMLR), "Stable and Efficient Gaussian Process Calculations", or Terenin et al. (2023, arXiv:2210.07893), "Numerically Stable Sparse Gaussian Processes via Minimum Separation using Cover Trees". Regularization through for instance inducing point choice helps, but at the cost of bias and performance. Our work suggests an orthogonal way to handle instability is to design algorithms that tolerate ill-conditioning well.
>
>
>
> 2. **Benign non-convergence**
>
> * > “That SGD converges slowly in extrapolation region is a bit concerning. Especially in applications like Bayesian optimization...high interest for exploration."
>
> * This is a very good comment; it illustrates why we find this work so exciting: a priori, we expected the same thing.
> Our empirical results instead showed that SGD can achieve strong performance in spite of non-convergence. In particular, SGD produces error bars in the extrapolation region which are closer to the prior than the true GP, and thus SGD overestimates uncertainty here. This may cause over-exploration, making convergence somewhat slower. We consider this "benign" compared to underestimating uncertainty which may cause catastrophic failure in Bayesian optimization (convergence to a local optimum).
>
> * The distribution-shift setting is hard to make reasoned arguments about, it is a rather ill-posed problem and there is no guarantee that the exact Bayesian model will perform best in such a setting.
>
>
>
>
> 3.  **Formal characterization of region-specific error.**
> A full mathematical characterization of the part of state-space where non-convergence occurs is what we initially aimed for; however, it proved too difficult. Such a characterization would require one to understand where in space do eigenfunctions corresponding to intermediate eigenvalues occur, which is non-trivial because it is a non-asymptotic question. We believe the amount of work required for this would warrant a separate submission. Alike the reviewer, we think the current analysis is insightful.
>
> 4. **The suggested extensions** are good ideas! *Non-conjugate inference* can be immediately achieved via the Laplace approximation as in Antorán et al. (2023), "Sampling-based inference for large linear models, with application to linearized Laplace". *Hyperparameter optimization* would require bi-level optimization: an outer loop for the hyperparameters and/or variational parameters, along with an inner loop for the linear systems. Since we cannot expect the inner loop to converge, one would need to study how to ensure that the outer loop behaves well even if the inner loop is not at the optimum.
>
> 5. **Questions on experiments**
>
> * You are correct:  **SVGP shares model hyperparameters with CG and SGD**, and also has some variational parameters where applicable.
>
> * We would expect **CG run for sufficiently long** to outperform all alternatives on all data sets. However, for houseelectric (2M+ points), this might take weeks. We are unable to commit that much compute.
>
>
> * **We do provide "RMSE to exact GP"** on our four small datasets in figures 3 and 8. In Table 1 we use test RMSE and NLL since for datasets with more than 50k observations (the focus of our work), we cannot do exact GP inference. In Table 1, for the four smallest datasets, CG converges to $10^{-2}$ tolerance and thus can be thought of as an **exact GP baseline**.
>
>
>
> 6. **Question: CG non-monotonicity**.
> This is a good question which we also asked ourselves for some time. Our best explanation: CG converges monotonically in the RKHS norm induced by the chosen kernel. For the Matérn kernel, this RKHS norm is (effectively, i.e. with certain parameter choices and up to norm equivalence) a weighted sum of the $L^2$ norms of the first $m$ derivatives plus the $L^2$ norm of the function itself. CG is thus trading off minimizing the norm of the $m$ derivatives of the function at the expense of the 0th order term: the $L^2$ error in the fit. The derivative norms being minimized first yields the divergence when looking at just $L^2$ error. Of course, since the RKHS norm is a sum of said $m+1$ non-negative terms, minimizing it will eventually force the 0th order term (error in $L^2$ norm) to go to zero too.
>
>
> 7. **Other**. Thanks for the suggestions! We have reworked Fig 4 and Proposition 1 to address your comments and add all required definitions.
>
> ----
>
> With these responses in mind, we gently and politely request that you please consider increasing your score towards firm acceptance.

---

> > ### Comment · Reviewer_Hc1f · 2023-08-17
> >
> > I want to thank the authors for their thoughtful response. Most of my questions are touched and addressed.
> >
> > However, my concern on the potential impact of this work is not fully resolved. In the authors response to Reviewer w7c5, two promising applications are (1) large-scale BO (with parallel Thompson sampling), and (2) large-scale spatio-temporal modeling. The paper investigates the first application in a synthetic setting (correct me if I am wrong). While the results seem encouraging, it would be more convincing to conduct the experiments in benchmark BO datasets (ideally also ill-conditioned to demonstrate its advantage). Is there a reason that the authors did not choose to do so? For (2), since the proposed method can only perform posterior sampling given fixed hyper-parameters instead of _learning_ the hyperparameters, how can it be useful for sptaio-temporal modeling? Is there a scenario where the large-scale posterior sampling is of interest in that domain?
> >
> >
> >
> > Again, I wanted to say that I believe the proposed method could potentially shine in many applications. However, I feel the paper and the authors response haven't really touched upon its real applicability. I would be happy to raise my score once this concern is addressed.

---

> > > ### Author Response · Authors · 2023-08-18
> > > **Real applicability: BO over molecular properties with Tanimoto kernel and more**
> > >
> > > Thank you very much for your reply! We go on to describe our ongoing work on the application of SGD GPs to molecular property prediction as well as how the method can be applied more generally, for instance, to spatiotemporal modeling.
> > >
> > > ----
> > >
> > > 1. We are currently working on **applying SGD inference to molecular binding energy prediction** using a dataset of 250k molecules introduced by [1]. We are using the Tanimoto kernel for graphs, which admits random features ([2]). In particular, we are searching for molecules which have a high probability of binding to proteins of interest using Bayesian Optimization.
> > >
> > > * The Tanimoto kernel only has 1 hyperparameter, the marginal kernel variance. For this task, the **authors of  [1] provide an optimized kernel hyperparameter** value, which they used in their experiments. Additionally, the **authors of [3] show how marginal kernel variances can be learnt using only GP posterior samples.** Thus, our SGD-based inference can be directly applied to this setting for learning the Tanimoto kernel’s hyperparameter.
> > >
> > > * Although there is not enough time to conclude these experiments before the end of the discussion period (the 21st), we would be happy to include them in the camera-ready version of the paper.
> > >
> > >
> > >
> > > 2. **We think that GP inference methods can be useful even without hyperparameter learning.**  A simple but general and effective approach to select hyperparameters is to **maximize the marginal likelihood on clustered subsets of the data**, as we do in our paper (See Appendix A.1). This yields results competitive with, and in some datasets outperforming, the hyperparameters learnt via conjugate gradients of [4]. This approach is particularly well-suited to length scale hyperparameters, which are of key importance in spatiotemporal modeling.
> > >
> > >
> > > 3. Next, **ill-conditioning appears consistently for large enough datasets or when the kernel distance between observations is small** (in fact, *often provably so* - see Section 2.3 of [5]), so one does not need to search particularly hard to find examples. The latter is bound to occur in Bayesian optimization, since methods often explore near previously-found well-performing locations.
> > >
> > > 4. Finally, there is strong precedent in the Gaussian processes where (a) **a novel method with significant advantages but important limitations was introduced**, and (b) **the limitations were addressed through follow-up work**.
> > >
> > > * For example, Titsias [6] introduced the variational-inference-based view of sparse Gaussian processes, developing a novel formalism for inducing points, whose complexity is $O(NM^2)$ - larger than for instance certain subset-of-data methods. Then, Hensman et al. [7] reduced this to $O(M^3)$ - a major improvement when $N$ is in the millions - by applying stochastic optimization to the variational inference objective. Achieving this improvement was only possible because the variational viewpoint had been developed previously.
> > >
> > > * Mirroring this example, **we expect that follow-up work, using for instance bilevel optimization techniques, can address limitations around hyperparameter learning** (for examples of such techniques in a neural network context, see [8,9]). This would start from the ideas we developed, but would likely introduce enough additional theoretical and methodological contributions, as well as experimental evaluation specific to hyperparameter optimization, to constitute another paper.
> > >
> > > [1] *DOCKSTRING: Easy Molecular Docking Yields Better Benchmarks for Ligand Design.*
> > > Miguel García-Ortegón*, Gregor N. C. Simm, Austin J. Tripp, José Miguel Hernández-Lobato, Andreas Bender, and Sergio Bacallado
> > >
> > > [2] *Tanimoto Random Features for Scalable Molecular Machine Learning*.
> > > Austin Tripp, Sergio Bacallado, Sukriti Singh, José Miguel Hernández-Lobato
> > >
> > > [3] *Sampling-based inference for large linear models, with application to linearised Laplace.*
> > > Javier Antorán, Shreyas Padhy, Riccardo Barbano, Eric Nalisnick, David Janz, José Miguel Hernández-Lobato
> > >
> > > [4] *Exact Gaussian Processes on a Million Data Points.*
> > > Ke Alexander Wang, Geoff Pleiss, Jacob R. Gardner, Stephen Tyree, Kilian Q. Weinberger, Andrew Gordon Wilson
> > >
> > > [5] *Numerically Stable Sparse Gaussian Processes via Minimum Separation using Cover Trees*.
> > > Alexander Terenin, David R. Burt, Artem Artemev, Seth Flaxman, Mark van der Wilk, Carl Edward Rasmussen, Hong Ge
> > >
> > > [6] *Variational learning of inducing variables in sparse Gaussian processes*. Michalis Titsias
> > >
> > > [7] *Gaussian processes for big data*. James Hensman, Nicolò Fusi, Neil Lawrence.
> > >
> > > [8] *Scalable One-Pass Optimisation of High-Dimensional Weight-Update Hyperparameters by Implicit Differentiation*. Ross M. Clarke, Elre T. Oldewage, José Miguel Hernández-Lobato
> > >
> > > [9] *Generalized Inner Loop Meta-Learning*. Edward Grefenstette, Brandon Amos, Denis Yarats, Phu Mon Htut, Artem Molchanov, Franziska Meier, Douwe Kiela, Kyunghyun Cho, Soumith Chintala

---

> > > > ### Comment · Reviewer_Hc1f · 2023-08-18
> > > >
> > > > Thanks to the authors for their response! These additional materials definitely strengthen the paper and would be great to be incorporated into the final version. I have raised my score from 6 to 7.

---

### Official Review · Reviewer_Hdii · 2023-07-07

**Soundness:** 4 excellent
**Presentation:** 4 excellent
**Contribution:** 4 excellent
**Rating:** 8
**Confidence:** 3

**Summary:**

This paper introduces a method for fast approximating a Gaussian process posterior when the data size is large. Exact computation complexity would be cubic in the data size, while this method is linear. It originates from the idea of pairwise conditioning of Gaussian process, where the law of Gaussian process posterior is expressed in terms of the law of Gaussian process prior. Decomposition into eigenfunctions gives a way of approximating the Gaussian process posterior, so the posterior inference transforms into appropriately choosing the coefficients of decomposition. Objectives are formed quadratic in the coefficients, which can be optimized with SGD for the sake of lower computational complexity compared to conjugate gradient methods. Ideas of inducing points are also discussed for reducing the data size needed. Intuitive discussion of errors in different regions are companied by figure illustrations as well as some theoretical results. Adequate numerical experiments are presented to support the method.

**Strengths:**

1. The paper has clear descriptions and is well written.
2. The ideas are mostly original and combine the advantages of multiple methods.
3. Many figure illustrations are present, making the ideas easily understood.
4. Supportive numerical experiments are conducted.
5. The problem of reducing the computational complexity of Gaussian process posteriors is itself very important and meaningful.

**Weaknesses:**

It would be nice to discuss why the Fourier basis are used for eigen decomposition and how it performs compared to other basis such as wavelets.

**Questions:**

To control the approximation error below a constant threshold, how does the number of basis L (number of components) needed scales with the data size and dimension? Either theoretical or numerical result could be interesting.

**Limitations:**

See Weaknesses and Questions.

---

> ### Author Rebuttal · Authors · 2023-08-08
>
>
> Thank you very much for your review! We are delighted that you found our descriptions “clear” and our paper “well-written” - thank you for these comments!
>
> > “It would be nice to discuss why the Fourier basis are used for eigen decomposition and how it performs compared to other basis such as wavelets.”* and asks the question *“To control the approximation error below a constant threshold, how does the number of basis L (number of components) needed scales with the data size and dimension? Either theoretical or numerical result could be interesting.”
>
> Thank you very much for these two closely-related questions! Due to the use of the terms “Fourier basis” and “spectral basis” in a non-synonymous way in our work, these questions can be interpreted in two ways: either (1) referring to the number of Fourier features, or to (2) the number of spectral basis functions along which we examine the convergence of SGD. Both questions are interesting, including potentially to other referees, so we will answer each of them.
>
> 1. **Fourier features.**
> In our work, Fourier features are used to (a) approximate the prior samples needed for pathwise conditioning, and (b) to approximate the regularization term $ || \alpha ||_K$  which appears in the quadratic objective used by SGD. Neither requires Fourier features explicitly: any finite basis function approximation of the prior kernel will work. None of our techniques are limited to Fourier features - we use them because they are convenient and work well for stationary kernels. For other kernels, including potentially non-stationary kernels, other bases such as wavelets or random hashes (Tripp et al. 2023, "Tanimoto Random Features for Scalable Molecular Machine Learning") could be used instead of Fourier features. This can be especially interesting for non-stationary kernels and kernels on boundary-constrained domains.
>
> * **On random (Fourier) feature approximation error**: We only use random features to approximate quantities that do not dependent on the targets: (a) prior function samples and (b) norms in the metric induced by the kernel matrix $\|\alpha\|_K$. In (a), the approximation error is also independent of the number of observations. We found this also to be the case empirically in (b).
>
> * Indeed, we use the same number of random features for across all experiments; 2000 for prior sampling and 100 for kernel matrix norms.
>
> * Crucially, we do *not* approximate any conditional distributions (i.e. matrix inverses) using random features; we use SGD for this. As the reviewer suggests, approximating conditionals with random features requires a number of features increasing in the number of observations and is prone to variance starvation. See Sutherland and Schneider (2015), "On the Error of Random Fourier Features" and Wilson et al. (2021), "Pathwise Conditioning of Gaussian Processes" for details including explicit error analysis.
>
>
>
>
> 2. **Spectral basis functions.**
> One can ask how many spectral basis functions one needs to look at in the convergence bound to ensure good performance. Recall that these are defined as $u^{(i)}(\cdot) = \sum_{j=1}^N U_{ji} / \sqrt{\lambda_i} k(x_j, \cdot)$, where $U$ is the U-matrix in the eigendecomposition of $K_{xx}$ and $\lambda_i$ are the respective eigenvalues. We chose the term “spectral” for these basis functions since they arise from the kernel matrix’s eigendecomposition. They define a “kernel-specific” basis that in some intuitive sense resembles the Fourier basis, but is different because it is data-dependent. As revealed by Proposition 1, the question of how many such basis functions are needed to ensure good performance is central to the performance of SGD in GPs, because it determines how many iterations are needed to ensure convergence. We do not address this question from a theoretical standpoint, because it is very technically difficult - please see our response to Referee Hc1f part (3) for more on this.

---

> > ### Comment · Reviewer_Hdii · 2023-08-21
> >
> > Thank you for the further explanations and responses to several questions. I am pretty satisfied with them and will keep the score as is.

---

### Author Rebuttal · Authors · 2023-08-09

We thank all reviewers for the time taken to read our paper and for their insightful and helpful comments. We are pleased the reviewers unanimously found our paper to be well-written, novel, and interesting.

----

The two most pressing concerns come from:

* Reviewers Hc1f and w7c5 ask about the **significance of the setting covered**.



The technique presented is immediately useful for **large-scale Bayesian optimization**. This problem appears most often in industrial settings and is the focus of our paper's second experiment

Our work also presents a novel approach to deal with **ill-conditioning in GP kernel matrices**. Ill-conditioned systems appear in almost all moderate-to-large-scale GP problems.

Further details on both of these points are given in the rebuttals for Reviewers Hc1f and w7c5.

----

* Reviewer xja3 is concerned about our **lack of comparison against Subset of Data (SoD) inference and our choice of SVGP hyperparameters**.

We have run **additional regression experiments using SoD methods**.

We further illustrate the strengths and weaknesses of SoD in **Figure 1(r) in the attached PDF**: SoD performs strongly only when the data is very redundant.
We have also **run the SVGP baseline with the number of inducing points increased to 4096.**

Quantitative results are provided in the individual response to xja3.

---

### Decision · Program_Chairs · 2023-09-21

**Decision:**

Accept (oral)

**Comment:**

This paper considers the use of stochastic gradient descent to approximately solve linear systems in Gaussian process (GP) modeling tasks, and this method is explored in a number of experiments for prediction and decision making. Overall, reviewers found the contribution to be novel and leading to interesting insights, and reviewers also appreciated presentation of the ideas and clean figures. Understanding a new approach of training GP models in large data settings is a topic of interest to a broad audience from researchers interested in methodology/theory to ones interested in applying scalable GP methods in practice.

We encourage the authors to revise the paper to incorporate suggestions from the reviewers and points made during the discussion phase, e.g., emphasizing the connection to broader areas where this work would be impactful, including the additional baselines presented during the rebuttal, and adding more in depth discussion of limitations and potential follow-up directions.